



# Source apportionment and impact of long-range transport on carbonaceous aerosol particles in Central Germany during HCCT-2010

Laurent Poulain[1], Benjamin Fahlbusch[1, *], Gerald Spindler[1], Konrad Müller[1], Dominik van Pinxteren[1], Zhijun Wu[1,**], Yoshiteru Iinuma[1,***], Wolfram Birmili[1,****], Alfred Wiedensohler[1], Hartmut Herrmann[1]

[1] Leibniz Institut für Troposphärenforschung, (TROPOS), Leipzig, 04318, Germany
[*] Now at Eurofins, Friedrichsdorf, 61381, Germany
[**] Now at College of Environment Sciences and Engineering, Peking University, Beijing, 100871, China
[***] Now at: Okinawa Institute of Science and Technology Graduate University (OIST), Okinawa, 904-0495, Japan
[****] Now at: Umweltbundesamt (UBA), Berlin, 14195, Germany

*Correspondence to*: Hartmut Herrmann (herrmann@tropos.de) and Laurent Poulain (poulain@tropos.de)

**Abstract.** The identification of different sources of the carbonaceous aerosol (organics and black carbon) was investigated at a mountain forest site located in central Germany from September to October 2010 to characterize incoming air masses during the "Hill Cap Cloud Thuringia 2010" (HCCT-2010) experiment. The near-$PM_1$ chemical composition, as measured by an Aerosol Mass Spectrometer (HR-ToF-AMS), was dominated by organics (OA, 41 %), followed by sulfate (19 %) and nitrate (18 %). Source apportionment of the OA fraction was performed using the Multilinear Engine approach (ME-2), resulting in the identification of five factors: Hydrocarbon-like OA (HOA, 3 % of OA mass), biomass burning OA (BBOA, 13 %), semi-volatile-like OA (SVOOA, 19 %), and two oxygenated OA (OOA) factors. The more-oxidized OOA (MO-OOA, 28 %) was interpreted as being influenced by aged polluted continental air masses, whereas the less-oxidized OOA (LO-OOA, 37 %) was found to be more linked to aged biogenic sources. Equivalent black carbon (eBC) measured by a multi-angle absorption photometer, MAAP, represented 10 % of the total PM. The eBC was clearly associated with the three factors HOA, BBOA, and MO-OOA (all together $R^2$ = 0.83). Therefore, eBC's contribution to each factor was achieved using a multi-linear regression model. More than half of the eBC (52 %) was associated with long-range transport (i.e. MO-OOA), whereas liquid fuel eBC (35 %) and biomass burning eBC (13 %) were associated with local emissions leading to a complete apportionment of the carbonaceous aerosol. The separation between local and transported eBC was well supported by the mass size distribution of elemental carbon (EC) from Berner-impactor samples.

Air masses with the strongest marine influence based on back trajectory analysis corresponded with a low particle mass concentration (6.4-7.5 µg m$^{-3}$) and organic fraction ($\approx$ 30 %). However, they also had the largest contribution of primary OA (HOA $\approx$ 4 % and BBOA 15-20 %), which was associated with local emissions. Continental air masses had the highest mass concentration (11.4-12.6 µg m$^{-3}$) and a larger fraction of oxygenated OA ($\approx$ 45 %) indicated highly processed OA. The present


results emphasize the key role played by long-range transport processes not only on the OA fraction but also on the eBC mass concentration and the importance of improving our knowledge on the identification of eBC sources.

**1 Introduction**

Atmospheric aerosol particles affect global climate through direct and indirect radiative forcing (IPCC, 2013), human health (Lelieveld et al., 2015;Burnett et al., 2014;Pope et al., 2011), as well as the ecosystems (Bohlmann et al., 2005;Jickells et al., 2005). Chemical composition of atmospheric particles at a specific sampling place (e.g. rural, urban, or marine environment) is not only depending on the local environment and sources but is also influenced by the history of the particles reaching the

sampling site. During transport, so-called aging processes not only modify the chemical composition of the particles but also affect their physical properties (e.g. size distribution, volatility, hygroscopicity, CCN activity, optical properties, Donahue et al., 2014;Farmer et al., 2015;Moise et al., 2015). As a consequence, aerosol particles at a specific location result from a complex mixture of different sources combined with complex processing.

Carbonaceous aerosol particles are a dominant fraction of total particle mass and are made of a large number of chemical species, which can be divided into organic aerosol (OA) and black carbon (BC) (e.g. Cabada et al., 2002). One of the most significant aerosol particle components influenced by atmospheric aging processes is the OA fraction, which can represent up to 90 % of the fine aerosol particle mass (e.g. Zhang et al., 2007). To better understand the origins of OA, source apportionment analysis is commonly applied to distinguish primary organic sources (e.g. related to fossil fuel, biomass, or coal combustion)

from secondary organic aerosol (SOA) sources based on either on-line measurements (e.g. Zhang et al., 2011;Canonaco et al., 2013), off-line chemical analysis (e.g. van Pinxteren et al., 2016;Srivastava et al., 2018) or a combination of both (Srivastava et al., 2019). Black carbon is associated with primary emissions from combustion processes of either anthropogenic (car, house heating, industry) or biogenic (e.g. wildfires) origins. In contrast to OA, source identification of BC remains sparse and only the recent development of an aethalometer model approach allows now to distinguish equivalent BC (eBC) related to traffic

emissions from wood combustion eBC (e.g. Sandradewi et al., 2008;Laborde et al., 2013;Zhu et al., 2018;Martinsson et al., 2017;Liakakou et al., 2020). Not only local sources drive the aerosol particle chemical composition; long-range transport, influenced by air mass origin, also plays an important role in local number size distribution and aerosol particle chemical composition (e.g. van Pinxteren et al., 2016;van Pinxteren et al., 2019;Waked et al., 2018). Therefore, not all the eBC mass concentration has to be linked to local sources, and a significant fraction can be attributed to long-range transport (e.g. Healy

et al., 2012;van Pinxteren et al., 2019).

The present work investigates the aerosol particle chemical composition and the different sources of carbonaceous particles reaching a site close to the village of Goldlauter in the Thuringian forest in Central Germany. The measurements were part of the "Hill Cap Cloud Thuringia 2010" (HCCT-2010) experiment, which aimed to investigate the impact of cloud processing to




aerosol physico-chemical properties. The Goldlauter site served as upwind site to study air masses before entering hill cap clouds at the Schmücke mountain. The present study focusses on an in-depth characterization at this site, while companion papers have related upwind site data to the other experiment sites (see https://www.atmos-chem-phys.net/special_issue287.html and e.g. Harris et al., 2013). The presented results stand for themselves, but can also be used to further interpret HCCT-2010 results and guide associated modeling and future experimental studies in the Schmücke area.

## 2 Site and instrumentations

### 2.1 Site

For the "Hill Cap Cloud Thuringia 2010" (HCCT-2010) experiment, the same places were used as for the FEBUKO/MODMEP experiments in 2001/02 (Herrmann et al., 2005). This work is focused on measurements performed at the upwind site (10∘45'20'' E, 50∘38'25'' N, 605 m a.s.l.) and approximately 6 km from the nearest city of Suhl. The sampling place is located

on the outskirts of the forest in front of the mountain site "Schmücke" under the prevailing SW wind direction and about 350 m below the mountain site "Schmücke" (see Tilgner et al. (2014) for more details). In the following, the upwind-site will be referred to as Goldlauter (GL). All times in the manuscript are given in Central European Summer Time (CEST).

### 2.2 Instrumentation

A large setup of on-line and off-line instruments was deployed during HCCT-2010, covering both gas and particle phases. On-

line instruments were sitting in two nearby laboratory containers and were operated continuously during the entire campaign. On the other hand, off-line sampling systems were applied only during specific intensive observation periods (IOPs) associated with two different conditions: 1- SW wind direction and presence of cloud at the Mt. Schmücke site for the full cloud events (FCE) and 2- SW or NE wind direction and no clouds or fog at any site for the non-cloud events (NCE). A detailed overview of these events is given by Tilgner et al. (2014).

### 2.2.1 Gas-phase measurements

The trace gases $O_3$, $SO_2$, and $NO_x$ (NO and $NO_2$) were quantified continuously using standard gas monitor systems for the whole duration of the experiment (Table SI-1). Non-methane hydrocarbon sampling (NMHC) was carried out with stainless steel containers every 2 hours during IOP events only. Ambient air was sucked into the container for three minutes before closing the valves. Analysis of the NMHC was carried out by the German Federal Environmental Protection Agency (UBA)

at their laboratory at Schmücke (UBA and GAW site) within 72 hours after sampling, using a GC-FID gas chromatographic analysis (Rappenglück et al., 2006). The cleaning of the stainless-steel container was carried out overnight through heating and evacuating.

Additionally, a Monitor for AeRosol and Gases in Ambient Air (MARGA 1S ADI 208, Metrohm AG, Switzerland, Rumsey et al., 2014;Twigg et al., 2015;Stieger et al., 2017 ) connected to a Teflon-coated $PM_{10}$ inlet was deployed for the detection of



the major water-soluble inorganic compounds in both gas phase and particle phase. A good correlation for the MARGA $SO_2$ and the UV-fluorescence monitor was reported in agreement with previous works from Makkonen et al. (2012);Stieger et al. (2017).

### 2.2.2 Aerosol particle measurements

#### 2.2.2.1 Online aerosol particle measurements

The online physico-chemical characterization of the ambient aerosol particles was performed using a large set of instruments. A High-Resolution Time-of-Flight Aerosol Mass Spectrometer (HR-ToF-AMS, Aerodyne Research Inc, DeCarlo et al. (2006), later referred to as AMS), a dual mobility particle size spectrometer (TROPOS-Type T-MPSS, Birmili et al., 1999), a Multi-Angle Absorption Photometer (MAAP, Model 5012, Thermo Scientific, Petzold and Schönlinner, 2004), a three-wavelength nephelometer (TSI Model 3563, Heintzenberg et al., 2006). All these instruments were located in the same laboratory container

and connected to the same sampling inlet consisting of a $PM_{10}$ inlet located approximately 6 m above ground level directly followed by an automatic aerosol diffusion dryer system maintaining the relative humidity in the sampling line below 30 % (Tuch et al., 2009). Moreover, water-soluble $PM_{10}$ inorganic ions were also measured by MARGA.

#### 2.2.2.2 AMS data analysis and positive matrix factorization

The AMS data was processed under the Squirrel version 1.52L and the PIKA version 1.13B (downloaded from http://cires.colorado.edu/jimenez-group/ToFAMSResources/ToFSoftware) using the IGOR Pro software package (Wavemetrics Inc, Portland, USA). A Chemical Dependent Collection Efficiency (CDCE) correction was applied on the AMS mass concentration according to Middlebrook et al. (2012). Quality assurance on the AMS data was achieved by comparing it to the MARGA and Berner impactors (sum of the first three stages) for individual species, while mass closure of the $PM_1$

aerosol particle chemical mass concentration as measured by the AMS and MAAP was achieved by comparing it to the estimated mass concentration from the T-MPSS and Berner impactors. Description and results can be found in sections 3.1.2 and 3.1.3 as well as in supplementary information SI-3 (AMS data validation, Fig. SI-1 to SI-4).

Source apportionment was performed on the high-resolution organic mass spectra dataset using the multi-linear engine (ME-2) model developed by Paatero (1999) and using the Source Finder tool (Sofi4.9, Canonaco et al., 2013) developed at the Paul

Scherrer Institute (PSI, Switzerland). Prior to analysis, the high-resolution organic mass spectra matrix was prepared according to the recommendations of Ulbrich et al. (2009). Isotope ions, which are calculated as a constant fraction of the parent ion, were removed. A minimum counting error was applied and ions with a signal-to-noise (SNR) ratio between 0.2 < SNR < 2 were down-weighted by a factor of 2, and ions with SNR < 0.2 were down-weighted by a factor of 10. Finally, ions related to $CO_2^+$ were also down-weighted since they are calculated as a fraction of the ion $CO_2^+$ (Allan et al., 2004). The source

apportionment was made following the recommendation of Crippa et al. (2014): in a first step, a non-constrained model approach was investigated. Since primary factors were not properly resolved during this first approach, a partially constrained approach was then investigated. Elemental analysis of the identified mass spectra was performed using the Analytic Procedure





for Elemental Separation (APES 1.06) based on Aiken et al. (2008) and including the improved approach from Canagaratna et al. (2015). A detailed description of the source apportionment analysis can be found in the supplementary information SI-5.


### 2.2.2.3 Off-line aerosol collection by 5-stage Berner-impactors and laboratory analysis

In parallel to the on-line measurements, a five-stage low-pressure Berner-impactor (LPI 80/0.05/2.9; Hauke GmbH und Co. KG, Austria, Berner and Lurzer, 1980) was used to collect PM size segregated during the IOPs using a humidity-controlled inlet (RH < 80 %). Water-soluble ions, organic carbon (OC), elemental carbon (EC), as well as sugar and anhydrosaccharide

analyses were performed. Details on the sampling conditions, sample preparation, as well as analytical methods can be found in the supplementary information (section SI-2). In the following, only results of IOPs longer than 2 h are considered in order to collect enough material on each stage of the impactor. Details on the sampling period for each considered IOP is provided in a dedicated section on the supplementary Information and in Table SI-2.

### 2.3 Back-trajectories and cluster calculations

The 96 h back trajectories were used to determine the influence of the air mass origin on aerosol. The trajectories were calculated for every hour from 13 September until 24 October 2010 for the altitude of 500 m above model ground with the NOAA Hybrid Single Particle Lagrangian Integrated Trajectory (HYSPLIT-4) Model (http://www.ready.noaa.gov/ready/hysplit4.html; Draxler and Hess, 2004) using the 1 degree resolution GDAS input data. The different back-trajectory clusters were calculated using the program R (http://www.r-project.org/; R Core Team, 2013) with

the package openair (http://www.openair-project.org; Carslaw and Ropkins, 2012;Ropkins and Carslaw, 2012).

## 3 Results

Data analysis will first focus on the overall aerosol particle chemical composition and mass closure. The second part will discuss the source apportionment of both organic aerosol and eBC. Finally, the third section will investigate the influence of the air mass origins on aerosol particle chemical composition and size distribution.

### 3.1 Aerosol particle chemical composition

### 3.1.1 Overall AMS-MAAP time series

Aerosol particle chemical composition (mass concentration and mass fraction) as measured by AMS and MAAP as well as the particle number size distribution over the entire time-period are shown in Figure 1. On average, the near-PM$_1$ particulate chemical composition was principally made-up of organic aerosol, OA (41 % of the total mass, Fig. 1). Sulfate and nitrate have quite similar contributions (19 % and 18 %, respectively).

have quite similar contributions (19 % and 18 %, respectively). The rest of the aerosol particle mass concentration was made of ammonium (11 %), eBC (10.0 %), and chloride (1 %). Despite their similar contribution to the particle mass fraction, sulfate and nitrate showed a clear time dependency (Fig. 1). Although sulfate dominates the inorganic fraction at the beginning of the



measurement period, nitrate becomes more important over time. This can be directly linked to a decrease of temperature during the sampling period (Fig. SI-5), inducing a change in nitrate partitioning between gas and particle phase. A last factor that

must be considered is the decrease of solar radiation from summer to winter, influencing the photochemical formation of sulfate. Variations of the organics and eBC mass concentration over the sampling period will be discussed in sections 3.2 and 3.3, respectively.

### 3.1.2 Berner impactor data

Parallel to online measurements, Berner-impactor provides size-resolved chemical composition up to 10 µm (Fig. 2). Over the

samples, $PM_{1.2}$ mass concentration (sum of the first three stages) represents more than 75 % of the $PM_{10}$. The only exceptions are for FCE 1.1, NCE 0.1, NCE 0.2, and NCE 0.3, which are all associated with back trajectory cluster C1-West and maritime air masses (see section 3.4). However, it is important to note that such aged maritime air masses do not systematically include an important super-µm fraction (for example, FCE 22.1). A simple reason for that is a washout of the air mass during rain events before it reaches the sampling site. A systematic look at the $PM_{1.2}$ to $PM_{10}$ ratio for the main aerosol component shows

that EC and sulfate are principally present on the $PM_{1.2}$ (> 80 % each), while $OC-PM_{1.2}$ represents more than 60 % of the $PM_{10}$ mass concentration. Nitrate is the only species that has a strong variability on the $PM_{1.2}$ to $PM_{10}$ ratio ranging from 20 to > 90 %).

### 3.1.3 AMS comparison with MARGA

According to the large contribution of the $PM_{1.2}$ mass concentration to $PM_{10}$ discussed above and despite their respective size

cutting, AMS and MARGA were successfully compared for ammonium, nitrate, and sulfate (Fig. 3, SI-2 and SI-3), indicating that these compounds were principally present in the $PM_1$ size range. In spite of the observed agreement, certain limitations must be considered. For example, the presence of organo-nitrates (ONs) and organo-sulfates (OSs) can interfere with the quantification of inorganic nitrate and sulfate by the AMS (Farmer et al., 2010;Bruns et al., 2010). The presence of such compounds might explain the slight overestimation of the AMS nitrate (Fig. 3) and sulfate (Fig. SI-2) happening from time to

time when they are compared to MARGA.

Additionally, some specific periods were also found in which the nitrate mass concentration measured by the MARGA appears to be higher and not related to the AMS's one (Fig. 3). This clearly indicates a larger contribution of super-µm particles that are not detected by the AMS as previously reported for the Berner-impactor samples. During these periods, MARGA's nitrate contains a larger fraction of sodium nitrate, which originates during the aging of marine aerosol when it crosses polluted areas

(e.g. Dasgupta et al., 2007), as confirmed by both the concomitant increase of sodium mass concentration, and the attribution of these periods to marine influence clusters (C1-West and C2-Northwest) with $RTI_{water}$ above 0.5 (Fig. 3 and further discussion in section 3.4). Stieger et al. (2018) drew similar conclusions by comparing the MARGA $PM_{10}$ measurements with an ACSM at the research site Melpitz in rural Germany. The very good results for the sulfate are also supported by that its size distribution inside the Berne-impactor samples.





### 3.1.4 Overall result of PM analysis


AMS, MARGA, and MAAP measurements provide complementary information on the aerosol particle chemical composition. Therefore, eBC from MAAP, organics from AMS, and inorganic ions from MARGA were combined to provide a comprehensive picture of the ambient $PM_{10}$ particle composition (Fig. 4) in a similar way as Schlag et al. (2016). The main advantage of combining these three instruments is to provide a more detailed description of the ambient particles than if they

were used individually. However, some limitations must be considered: i) crustal material is detected neither by MARGA nor AMS and therefore will not be considered at all in the following discussion; ii) the use of different upper-size cutting for the OA might lead to an underestimation of the total OA mass as expected from the OC distribution. However, the resulting $PM_{10}$ estimation correlates very well with the $PM_{10}$ mass concentration measured by the Berner-impactor (slope of 0.98, $R^2$ = 0.96, Fig. 4), indicating a minor contribution of non-considered species (i.e. dust, calcium, magnesium, and trace metals) to the total

$PM_{10}$ mass, as well as a lower organic contribution to the super-µm size range. Although the estimated $PM_{10}$ mass was validated only during the IOPs, it appears to be reasonable to consider it as accurate over the entire experiment within the previously mentioned limitations.

### 3.2 Organic aerosol source apportionment

An investigation of the organic aerosol source apportionment highlights the presence of five different factors, which were

identified based on their individual time series, mass spectra, diurnal variability, and comparison with external measurements (Fig. 5). A detailed description of the different steps of the analysis as well as the identification of the different factors is given in the supplementary information (Section SI-5). Briefly, in a first step, a non-constrained model was run, and, in a second step, a series of partly constrained runs were investigated in order to better distinguish the different primary organic factors. The selected final solution results in a partially constrained model with two primary organic factors: Hydrocarbon-like organic

aerosol (HOA) and biomass burning organic aerosol (BBOA). HOA was constrained using the mass spectra reported by Mohr et al. (2012) in Barcelona (Spain) and is available on the AMS mass spectra database (http://cires.colorado.edu/jimenez-group/HRAMSsd/, Ulbrich et al., 2009). For BBOA, the mass spectra identified during the source analysis process itself was used (see details in section SI-5). In addition to these two primary OA factors, three distinct oxygenated organic aerosols (OOA) were identified as semi-volatile OOA (SV-OOA), low-oxidized OOA (LO-OOA) and more-oxidized OOA (MO-OOA)

(Fig. 5). It is important to note that the three OOA factors were already identified in the first step of the source apportionment analysis during the non-constrained approach.

### 3.2.1 HOA

HOA is commonly considered a surrogate for fossil fuel combustion emissions, especially related to traffic emissions. The HOA mass spectrum is characterized by a larger contribution of hydrocarbon-like ions ($C_xH_y^+$ fragments, Fig. 5), resulting in

a low O:C (0.04) and high H:C (2.00), which is in agreement with previously reported values from Canagaratna et al. (2015).





HOA correlates reasonably with *trans*-2-pentene ($R^2 = 0.43$), *cis*-2-pentene ($R^2 = 0.56$), eBC ($R^2 = 0.45$), benzene ($R^2 = 0.62$), toluene ($R^2 = 0.35$), $NO_2$ ($R^2 = 0.31$), and CO ($R^2 = 0.25$) (Fig. SI-16). Moreover, eBC, CO, and $NO_x$ have similar diurnal patterns as HOA, with two maxima (early in the morning (09:00) and early in the evening (18:00), Fig. SI-17), which is typical for car emissions and/or house heating using fossil fuel. Correlation between HOA and eBC must be carefully interpreted,

since periods with similar trends alternate with periods of very different covariance (Fig. 5). This clearly indicates the presence of other eBC related sources, as will be discussed later on. On average, over the whole period, HOA contributed $3 \pm 3$ % (mean $\pm$ standard deviation) of the total organic mass concentration, designating HOA as a minor source of OA at the sampling place. A clear dependency of HOA contribution on OA and temperature was observed (Fig. 6), ranging from around 1 % during the warmest period up to 5 % for the coldest period. This temperature dependency indicates that HOA should be mostly associated

with local residential house heating rather than car exhaust.

### 3.2.2 BBOA

The BBOA factor ($13.3 \pm 10.0$ % of the total OA) is related to biomass burning emissions and its mass spectrum is characterized by the presence of two specific fragments: $C_2H_4O_2^+$ at m/z 60 and $C_3H_5O_2^+$ at m/z 73 (Fig. 5); they are known to be related to anhydrous sugars like levoglucosan (e.g. Schneider et al., 2006). This is confirmed by the correlation observed in off-line

levoglucosan ($R^2 = 0.84$, Fig. SI-16). The elemental ratios of the BBOA factor correspond with the lower range of all the reported values in Canagaratna et al. (2015). Although BBOA mass spectrum is influenced by the type of wood used, the combustion conditions, as well as the wetness of the wood (Ortega et al., 2013), the low O:C value of the BBOA factor indicates that it is mostly freshly emitted. BBOA also correlates with specific NMHC compounds like ethane ($R^2 = 0.34$), ethylene ($R^2 = 0.55$), *m,p*-xylene ($R^2 = 0.32$), and more generally with the total alkenes ($R^2 = 0.40$), as previously reported by

Gaeggeler et al. (2008) and Schauer and Cass (2000). Similarly to HOA, the BBOA fraction to total OA increases with the decrease of the temperature, representing almost 20 % of the OA during the coldest period (Fig. 6). The correlation between BBOA and HOA ($R^2 = 0.86$) clearly indicates that these two factors are emitted from a similar origin. This supports our conclusions on the HOA as being mostly associated with residential house heating.

Wood combustion used for residential house heating dominates the local anthropogenic emissions in the surrounding area of

the sampling place. This is in agreement with the reported BBOA contribution of 20 % for a similar place in Germany in winter (Poulain et al., 2011). The predominance of biomass burning emissions compared to liquid fuel is also supported by the benzene to toluene ratio value during the IOPs (mean: 1.1, min.: 0.47, max.:2.65), which is comparable to the ratio reported by Gaeggeler et al. (2008) for a similar location in Switzerland.

Overall, the sum of the primary OA (POA = HOA + BBOA) were principally associated with back-trajectory clusters C1-West and C2-Northwest (20-25 % of OA) (see section 3.4), while they only contribute < 11 % to the other clusters. Consequently, taking all of them together, the C1-West and C2-Northwest clusters might be impacted by the village of Goldlauter, as discussed later in section 3.4.



### 3.2.3 SV-OOA

The SV-OOA was identified according to the relative similarity of its time series with nitrate ($R^2 = 0.40$) and its diurnal profile, which shows the highest concentrations during nighttime, a decrease in the early morning hours, and a minimum during daytime (Fig. SI-17). The SV-OOA mass spectrum is characterized by a higher contribution of the ions $C_2H_3O^+$ (m/z 43) compared to $CO_2^+$ (m/z 44). The elemental analysis of the SV-OOA mass spectra shows an O:C of 0.40 and a H:C of 1.70 (Fig. 5), which is in the lowest range of the reported SV-OOA values from Canagaratna et al. (2015). Its MS is also similar to

the median SV-OOA mass spectra based on 25 AMS measurements over Europe ($R^2 = 0.69$) described by Crippa et al. (2014). The difference was mainly attributed to a smaller contribution of the $CO_2^+$ ion (m/z 44) on our factor. On average, throughout the entire sampling period, SV-OOA represented $19 \pm 11$ % of the total OA, ranging from 12 to 22 %, with a clear temperature dependency similar to the POAs (Fig. 6). Nevertheless, this is in agreement with the fact that SV-OOA is generally associated with gas-to-particle partitioning of semi-volatile organic compounds (*e.g.* Ulbrich et al., 2009;Zhang et al., 2011).

Consequently, it has to be related to a more local and /or regional influence. Interestingly, SV-OOA also correlates with anthropogenic NMHC gases like toluene ($R^2 = 0.55$), *i*-pentane ($R^2 = 0.52$), benzene ($R^2 = 0.46$), as well as the sum of aromatics NMHC ($R^2 = 0.60$) (Fig. SI-18). This correlation may be a consequence of the presence of semi-volatile organic compounds and intermediate volatility organic compounds (VOCs and IVOCs, respectively) either directly emitted from anthropogenic sources or resulting from the dilution of the POA (e.g. Lipsky and Robinson, 2006;May et al., 2013). Ambient relative humidity

and, subsequently, aerosol liquid water content seem to be two of the essential parameters driving the partitioning of the anthropogenic VOCs and IVOCs in the particle phase (Murphy et al., 2017). Therefore, correlations between SV-OOA and NMHC might be results from the condensation of semi-volatile compounds emitted by anthropogenic sources.

### 3.2.4 OOAs

Finally, the two OOAs referred to as low-oxidized oxygenated organic aerosol (LO-OOA) and more-oxidized oxygenated

organic aerosol (MO-OOA) were identified during an early stage of the source apportionment analysis, as discussed in the supplementary information (section SI-5). They present two distinct time series and mass spectra, indicating two different sources rather than an artificial splitting by the model (Fig. 5). Both are characterized by a high contribution of mass m/z 44 (mostly $CO_2^+$), while only LO-OOA has a strong contribution of m/z 43 (mostly $C_2H_3O^+$). Their elemental ratios reflect this difference. MO-OOA is more oxygenated (O:C = 0.89) than LO-OOA (O:C = 0.58). The difference between the two OOAs

might be related to either different precursors or aging processes.

The two OOAs are the two most important contributors to the total OA fraction ($28 \pm 12$ % and $37 \pm 18$ % of the OA for MO-OOA and LO-OOA, respectively). However, their individual relative contributions strongly vary over time. LO-OOA dominated at the beginning of the measurement period and contributed up to ca. 60 % of OA (Fig. 5). The fragment $C_7H_7^+$ (m/z 91) was frequently associated with biogenic SOA, even though it cannot be considered as a specific tracer (Lee et al.,

2016). $C_7H_7^+$ arises from the fragmentation of aromatic compounds and can, therefore, have several sources. Here, the





contribution of $C_7H_7^+$ to total OA ($f_{91}$) is higher in LO-OOA than MO-OOA, which might indicate a larger contribution of biogenic SOA to LO-OOA. Opposite to the anthropogenic related factors, the mass fraction of LO-OOA decreases with the decrease of temperature (Fig. 6).

This can be associated with the decrease of the biogenic VOC emissions from late summer to early winter (Helmig et al., 2013). The impact of biogenic sources is also supported by the air mass cluster analysis, which associated the highest fraction of LO-OOA with a cluster with the highest RIT over natural vegetation (Table 1 and discussion in section 3.4).

In contrast, MO-OOA does not show a pronounced temperature dependency, but it strongly correlates with eBC ($R^2 = 0.79$, Fig. 5 and SI-16), which is higher than the coefficient correlation for the POA factors. MO-OOA also correlates better with

oxalic acid ($R^2 = 0.81$) than LO-OOA ($R^2 = 0.65$) (not shown). Moreover, MO-OOA correlates moderately with $SO_2$ ($R^2 = 0.35$), suggesting an anthropogenic influence. It is known that the aging of primary OA leads to mass spectra with a similar pattern to OOA (Jimenez et al., 2009). Consequently, MO-OOA can be identified as being related to processed polluted/anthropogenic air masses from long-range transport.

**3.3 Equivalent black carbon (eBC) source apportionment**

As mentioned before, eBC correlated with three different organic factors (HOA, BBOA, and MO-OOA) identified during source apportionment analysis. Taken together, the sum of these factors correlates strongly with eBC ($R^2 = 0.83$) as shown in Figure 7-a. Therefore, using a multilinear regression model, the different sources of eBC were assessed for each factor following Laborde et al. (2013) and Zhu et al. (2018). The assumption made here is that the eBC mass is attributed to individual contribution of each OA factor (i.e. $eBC_{HOA}$, $eBC_{BBOA}$, and $eBC_{MO-OOA}$) at any time as following:


$$eBC(t) = eBC_{HOA}(t) + eBC_{BBOA}(t) + eBC_{MO-OOA}(t) \qquad (1)$$

The eBC emission related to each source is assumed to be proportional to the individual source mass concentration released ($m_{HOA}$, $m_{BBOA}$, and $m_{MO-OOA}$, respectively). Therefore, the multilinear regression model could be explained as follows:


$$eBC(t) = a\, m_{HOA}(t) + b\, m_{BBOA}(t) + c\, m_{MO-OOA}(t) \qquad (2)$$

Where $a$, $b$, and $c$ respectively represent the linear regression coefficient for $m_{HOA}$, $m_{BBOA}$, and $m_{MO-OOA}$, which will be used to estimate the respective eBC contribution towards each OA factor.

A very good correlation between measured and modeled eBC was obtained (Figure 7-b), and modeled eBC explained 96 % of the measured one. Based on this approach, long-range transport particles associated with MO-OOA are the largest source of eBC during the measurement period, contributing to half of it (52 %), while eBC associated with local emissions of HOA and BBOA represents 35 % and 13 %, respectively. Considering only local eBC sources, fossil fuel combustion dominates the





eBC fraction (73 % for eBC$_{HOA}$ and 27 % for eBC$_{BBOA}$), which in agreement with previous works (e.g. Healy et al.,
2012;Herich et al., 2011).

Using single-particle mass spectrometer measurements, Healy et al. (2012) reported that size distribution of EC can also
directly be used to apportion soot sources: a local EC source was related to particles with a vacuum aerodynamic diameter
(d$_{va}$) of < 400 nm, while continental transported soot was related to particles with d$_{va}$ > of 400 nm. A similar cut-off diameter
was applied to the Berner-impactor measurements to split the EC into freshly emitted and transported one, assuming that the
first two stages (i.e. aerodynamic diameter ranging from 50 to 420 nm) were associated with the local EC, and the three larger
ones (i.e. diameter ranging from 420 nm to 10 µm) with continental transport (Fig. 8). The resulting EC classification provides
quite similar results than the one discussed before and using the multilinear regression approach (Fig. 8) with an uncertainty
of approximately 20 % (mean ratio between the two approaches excluding FCE 11.2) supporting our conclusions. Only FCE
11.2 provides completely different results between the two approaches without any clear explanation. In summary,
comparisons between the two approaches (multilinear regression and impactor size cutting) support each other and both
confirm the importance of a minimum near 400 nm (in aerodynamic diameter) in EC size distribution for distinguishing freshly
emitted from long-range transported soot. The importance of long-range continental transport of soot is also in agreement with
the measurements made by Roth et al. (2016) at the summit station by the ALABAMA. The authors reported that soot was
mainly found in particles with diameters larger than 450 nm, which correspond with aged/processed soot.

### 3.4 Influence of air mass origin to chemical composition and particle number size distribution

A total of 6 clusters was obtained based on 96 h backward air mass trajectories (Fig. 9) and they are characterized in Table 1
by their residence time index (RTI) over different types of ground before reaching the sampling place, based on the approach
described in van Pinxteren et al. (2010), and their meteorological conditions. Cluster C1-West and C2-Northwest correspond
to two different types of marine-influenced air masses with RTI$_{water}$ of 0.34 and 0.47, respectively. C1-West starts near Iceland,
while C2-Northwest comes from the Norwegian Sea. These trajectories occurred during 31 % and 17 %, respectively, of the
measurement period. Although the cluster C3-Southwest (18 % of the time) also contains a maritime component at the starting
point of the air masses (RTI$_{water}$ = 0.12), it is dominated by an RTI associated with continental areas (France and South-
Germany). Cluster C4-South is characteristic of southern Europe, coming from an industrial and polluted area of northern
Italia via Austria and the south of Germany. It also presents the highest RTI (0.51) related to natural vegetation (i.e. forest).
Furthermore, it corresponds to the warm period. However, this cluster only occurs for a short period (8 % of the total sampling
period). Cluster C5-Northeast is an issue of the boreal area (north of Sweden), spending some time over the Baltic Sea (near
Finnish, Latvian, and Polish coasts) before entering the north of Germany. Although it might also contain a small maritime
component (RTI$_{water}$ = 0.18), this cluster mostly follows coastal areas. Therefore, it should present an important continental
and polluted aspect. Similar to C4-South, it is not very common (9 % of the sampling period). Finally, C6-East clearly
represents continental air masses coming from the east side of Europe (crossing Russia, Ukraine, Poland, and the Czech





Republic). The sampling site was under its influence 17 % of the time, mostly during the second part of the experiment. Not surprisingly, the air mass clusters with the highest continental background (C3-Southwest, C4-South, and C6-East) also correspond with the ones with the highest aerosol particle mass concentrations (Table 1).


The aerosol particle chemical composition, number size distribution (PNSD), and trace gases were averaged according to the different air mass clusters and are presented in Figure 10, and summarized in the supplementary information (Table SI-4). The highest particle mass concentrations were observed for clusters with the strongest continental influence (i.e. C3-Southwest, 11.5 µg m$^{-3}$; C4-South, 11.4 µg m$^{-3}$, and C6-East 12.6 µg m$^{-3}$). The largest mass concentrations of chloride, sodium, and

potassium salts were associated with C1-West and C2-Northwest, which have the largest marine influence (Fig. 10 and Table 1). These two clusters also correspond to the periods with the lowest particle mass (7.5 µg m$^{-3}$ and 6.4 µg m$^{-3}$, respectively) and trace gas concentrations (Table SI-4). Although the lowest absolute and relative mass concentrations of organics ($\approx$ 30 %) and eBC (7-8 %) were also observed for these two clusters, they show the largest fraction of anthropogenic sources (HOA $\approx$ 4 % and BBOA 15-20 % of OA). The average particle number size distributions for these two clusters present the highest

concentration of Aitken mode particles (centering on 40-50 nm), supporting the influence of local anthropogenic emissions. Increasing the RTI value of the air masses over continental areas leads to an increase of the carbonaceous fraction in both absolute and relative mass concentrations. The highest mass concentrations of OA and eBC are associated with C4-South (5.4 µg m$^{-3}$ and 1.0 µg m$^{-3}$, respectively) and C6-East (5.8 µg m$^{-3}$ and 1.3 µg m$^{-3}$, respectively). The averaged particle size distribution for C3-Southwest and C4-South are almost unimodal, centering on 100 nm, indicating well-processed particles

not impacted by local anthropogenic sources. The highest mass concentrations of eBC and OA are both linked to air masses coming from eastern Europe (C6-East). This is also true in regard to gas composition since the cluster C6-East also shows the highest concentrations of SO$_2$, HNO$_3$, and O$_3$ (Table SI-4). Interestingly, the different nucleation events observed during the sampling period (Wu et al., 2015) were mainly connected to clusters C5-Northeast and C6-East, as reflected by the large contribution of the smallest particle diameters (nucleation mode) to their respective PNSD profiles, which did not appear on

the other clusters. Nucleation processes strongly depend on the sulfuric acid formation (Kulmala et al., 2004;Birmili et al., 2003), which is certainly promoted by the SO$_2$ concentration. Finally, the two continental eastern clusters also show a large particle mode (approx. 200 nm), indicating long-range processed particles in agreement with the large fraction of MO-OOA and its associated eBC. Taken together, the aerosol particle and trace gas compositions of C6-East confirm the strong anthropogenic influence on air masses originating from this sector, which is in agreement with long-term measurements

performed at the German station of Melpitz located approximately 250 km to the north-east of the Goldlauter site (Spindler et al., 2013).



## 4 Summary and Conclusion

In the frame of the HCCT-2010 campaign, a detailed description of the aerosol particle chemical composition reaching the site of Goldlauter was made by combining continuous online measurements (AMS, MARGA, MAAP, and MPSS) with offline
impactor samples performed during specific IOPs. Merging online results from the AMS, MAAP, and MARGA together provides an hourly time-resolved chemical picture of the ambient $PM_{10}$ composition. The consistency with the merged total $PM_{10}$ mass and the one measured by the Berner-impactor highlight the fact that OA on the super-µm size range should be low, and non-considered species (i.e. dust, Calcium, Magnesium, and trace metals) should play a minor role in the total $PM_{10}$ mass during our study. A total of five factors were identified from the source apportionment of the OA, including two primary
organic aerosols related to fossil fuel combustion, HOA (3 % of total OA), biomass burning combustion BBOA (13 %), and three oxygenated organic aerosols including a semi-volatile SV-OOA (19 %), a MO-OOA (28 %) associated with long-range transport of polluted continental air masses, and a LO-OOA (37 %) related to aged biogenic aerosol. Using the correlation between HOA, BBOA, and MO-OOA with eBC, a multilinear regression approach was applied to performed source apportionment of eBC. This analysis highlights eBC contributions related to the source of HOA (35 %), BBOA (13 %), and
MO-OOA (52 %). It was therefore possible to distinguish local eBC emissions (48 % of the total eBC) dominated by fossil fuel combustion (73 % of the local eBC) from long-range transported eBC. Size resolved EC from the Berner-impactor supports our founding and confirms the applicability of the size cutting at 400 nm for distinguishing local EC (< 400 nm) from transported EC (> 400 nm) as mentioned by Healy et al. (2012). Local sources of organic aerosol, appear to play a smaller role compared to long-range transport, which is responsible for 65 % of OA. These results confirm the importance of long-range
transport to the total mass concentration of OA and eBC not only during high pollution events (e.g. van Pinxteren et al., 2019;Petit et al., 2017;Waked et al., 2018). HOA and BBOA are properly depicted by the primary anthropogenic sources of liquid fuel and biomass burning emissions, respectively. However, during their transport, aging processes will lead to signatures closer to OOA (Jimenez et al., 2009). A direct consequence is that the MO-OOA associated with long-range transport of polluted air masses, might result from such aging processes of anthropogenic sources and it highlights the
complexity of the internal chemical composition of the OOAs. Consequently, further efforts should be made in the future on the improvement of the OOAs' identification in order to properly distinguish between biogenic-SOA and anthropogenic-SOA, as well as between locally formed and transported SOA, which could be critical for rural and remote stations.

*Data availability*: All data is available upon request to the corresponding author.
*Authors contributions*: LP, FB, GS, KM, DvP, ZW, YI, collected the data, LP performed data analysis on the AMS, FB, GS, KM, DvP, ZW, YI contributed to the evaluation of the off-line chemical analysis dataset and WB evaluated the MPSS dataset. All co-authors participated in the interpretation of the results. LP lead the writing of the manuscript to which all authors contributed.




*Competing of interest*: the authors declare that they have no conflict of interest.

**Acknowledgements**

This work was supported by the German Research Foundation (DFG) under He 3086/15-1.

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



**Table 1: Properties of the different air mass clusters**

| | Cluster 1 | Cluster 2 | Cluster 3 | Cluster 4 | Cluster 5 | Cluster 6 |
|---|---|---|---|---|---|---|
| Air mass origin | West | Northwest | Southwest | South | Northeast | East |
| Frequency (%) | 31 | 17 | 18 | 8 | 9 | 17 |
| Average Residence Time Index (RTI) | | | | | | |
| Water | 0.34 | 0.47 | 0.12 | 0.6 | 0.18 | 0.4 |
| Agriculture area | 0.40 | 0.31 | 0.41 | 0.41 | 0.46 | 0.59 |
| Natural vegetation | 0.24 | 0.19 | 0.44 | 0.51 | 0.33 | 0.35 |
| Urban area | 0.2 | 0.2 | 0.1 | 0.1 | 0.2 | 0.2 |
| Other global properties | | | | | | |
| Average length (km) | 3111 | 3438 | 2141 | 1716 | 2514 | 2561 |
| Average solar radiation ($W\ m^{-2}$) | 133 | 103 | 138 | 155 | 96 | 137 |
| Average condition at the sampling site | | | | | | |
| Temperature (°C) | 7.7 | 5.6 | 11.3 | 22.8 | 6.2 | 7.4 |
| RH (%) | 91.6 | 92.7 | 89.6 | 90.5 | 88.2 | 72.0 |
| Precipitation (mm) | 0.30 | 0.16 | 0.18 | 0.08 | 0.08 | 0.17 |
| Wind speed ($m\ s^{-1}$) | 4.21 | 3.95 | 3.84 | 3.67 | 3.41 | 3.58 |
| Particle mass concentration ($\mu g\ m^{-3}$) | 7.54 | 6.38 | 11.50 | 11.38 | 7.97 | 12.64 |



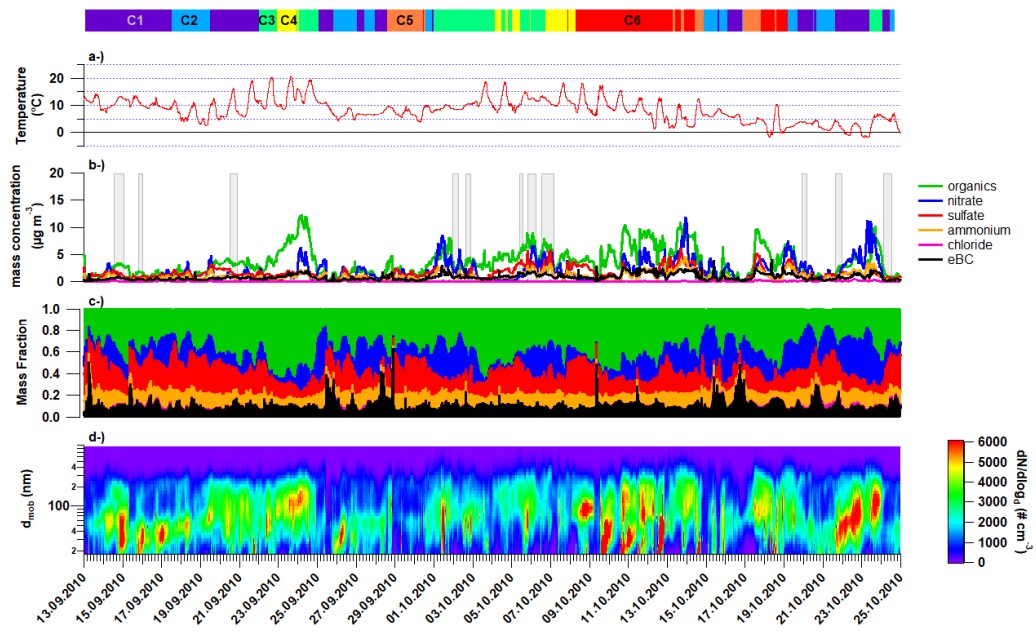

**Figure 1:** Time series of the ambient temperature (a), the particulate near-PM₁ chemical composition as measured by the AMS and completed by MAAP for equivalent black carbon (b), the corresponding mass fraction (c), and particle number size distribution (d) during HCCT-2010 at the site of Goldlauter. The colored bars and numbers at the top refer to the six different air mass clusters (see section 3.4), and the grey bars refer to the different cloud and non-cloud events as defined in Table SI-2.

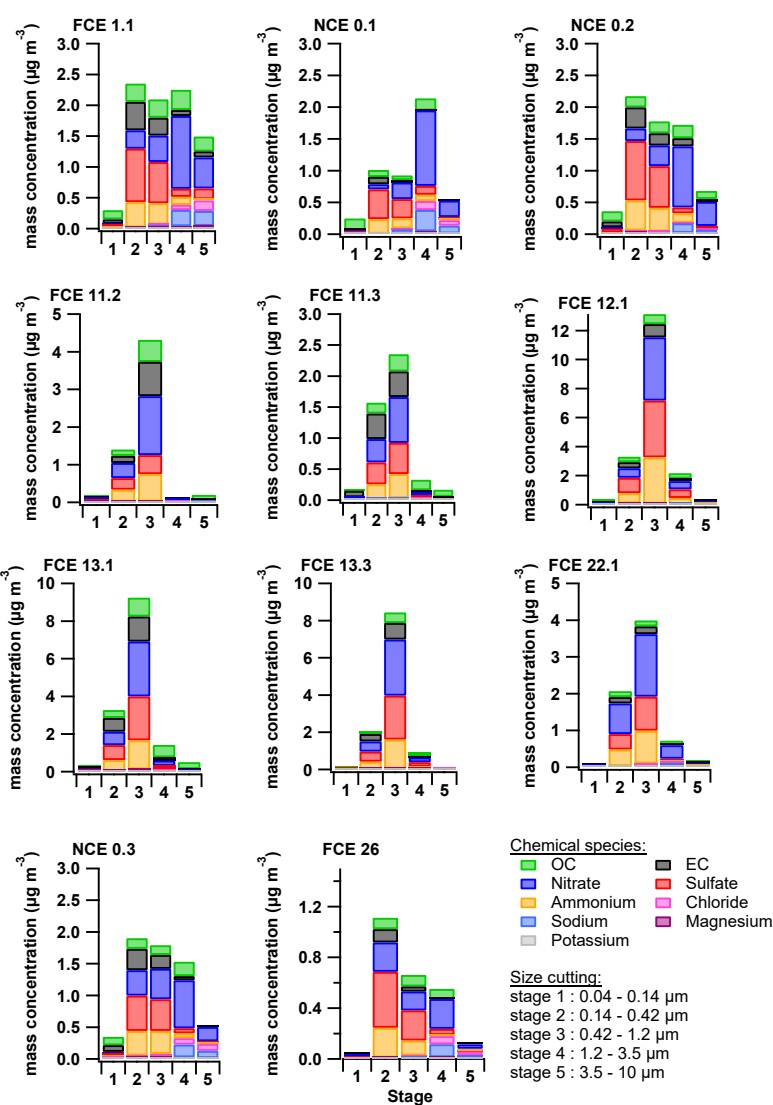

**Figure 2: Size distribution of OC, EC, and major water-soluble ions from Berner-impactor measurements for the different events.**



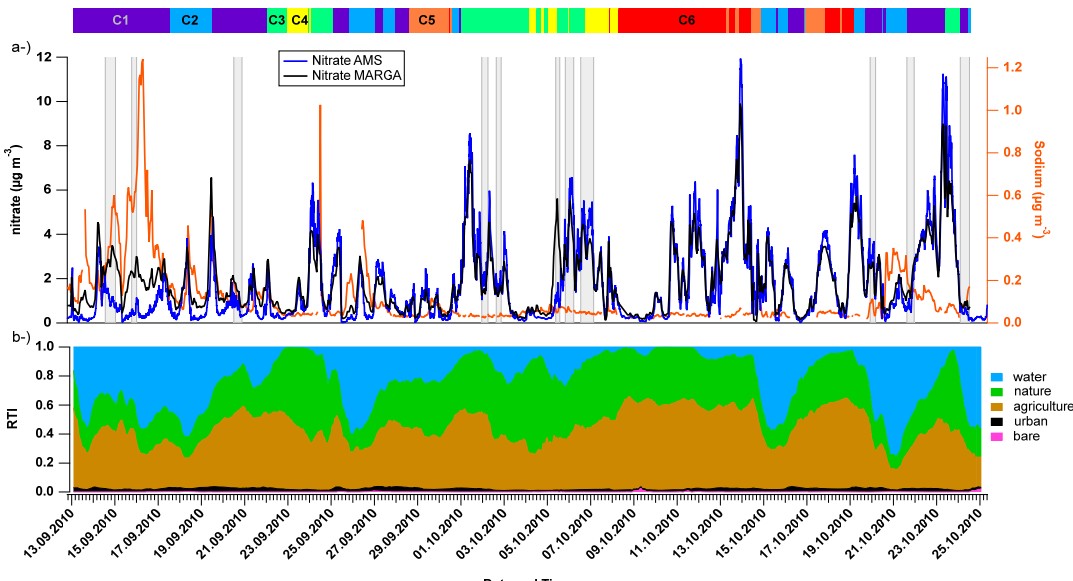


**Figure 3: Influence of marine air masses on nitrate size distribution: (a) nitrate mass concentrations measured by AMS and MARGA as well as sodium mass concentrations by MARGA, (b) the residence time index (RTI) of the 96 h backward trajectories above different land cover as developed by van Pinxteren et al. (2010). The colored bars and numbers at the top refer to the 6 different air mass clusters (see section 3.4), and the grey bars refer to the different cloud and non-cloud events as defined in Table SI-2.**


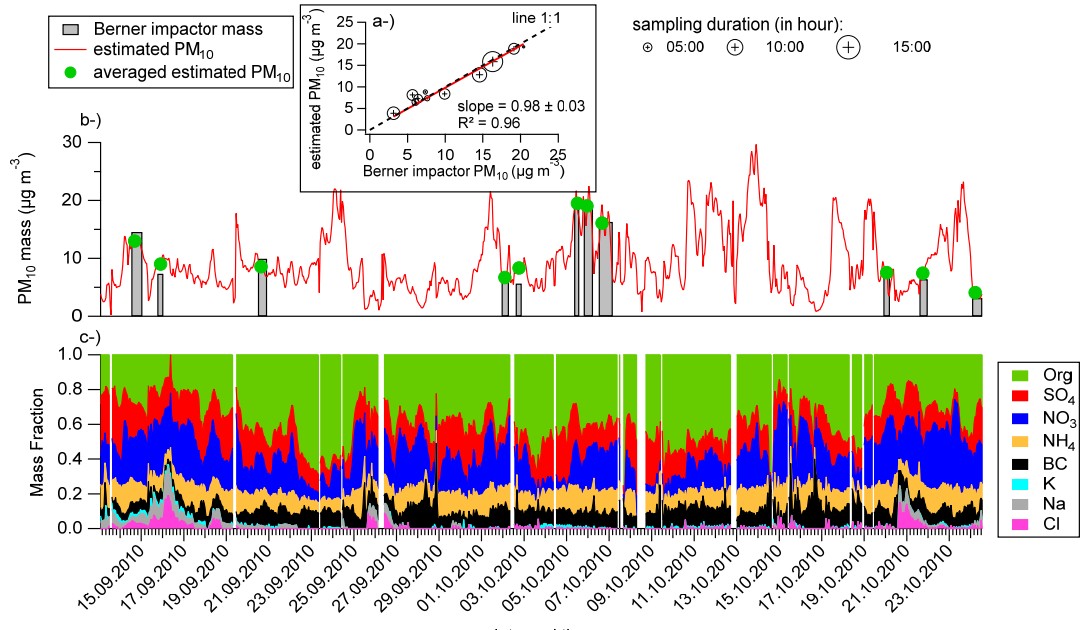


**Figure 4: Estimation of the PM₁₀ mass concentration by combining AMS, MAAP, and MARGA: (a) Comparison with total PM₁₀ mass concentration measured by Berner-impactors during intensive sampling periods, (b) mass concentration time series, (c) mass fraction of the main components. The size of the point on the scatter plot (a) refers to the sampling duration of the Berner-impactor. The red line in the insert panel represents regression fit by least orthogonal distance fit forced to zero.**


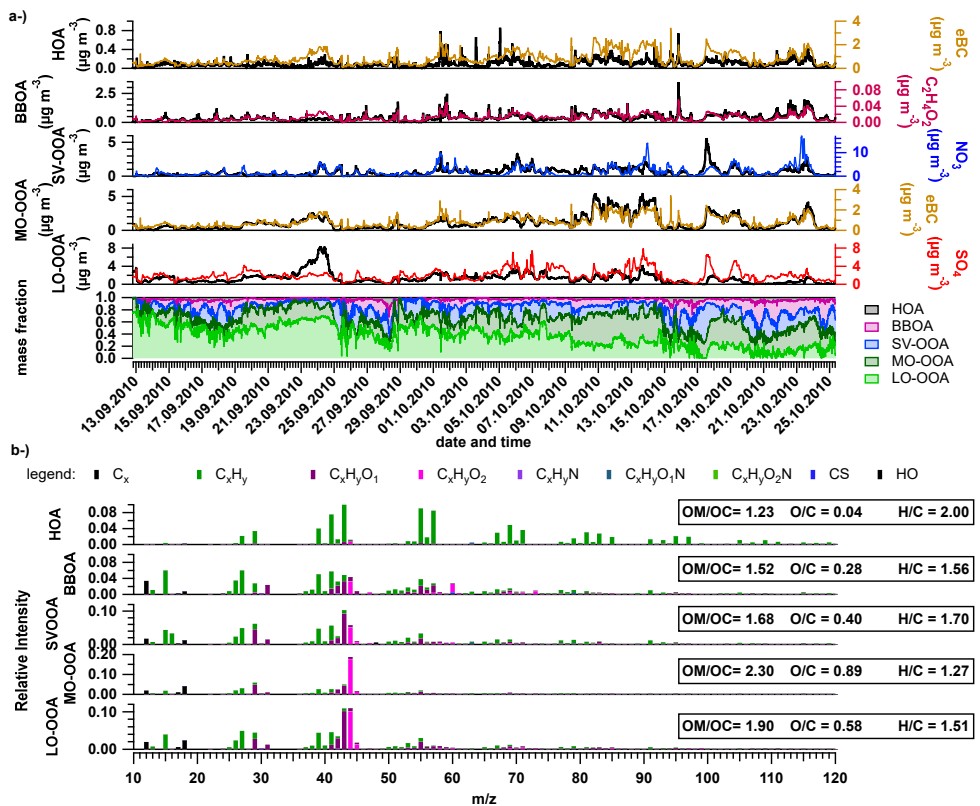

**Figure 5: Overview of the ME-2 results. a) Time series of the 5-factors solution mass concentration and their corresponding tracers as well as mass fraction of each factor throughout the experiment. b) High-resolution mass spectra colored by the different groups of fragments and the elemental ratios of the identified factors.**





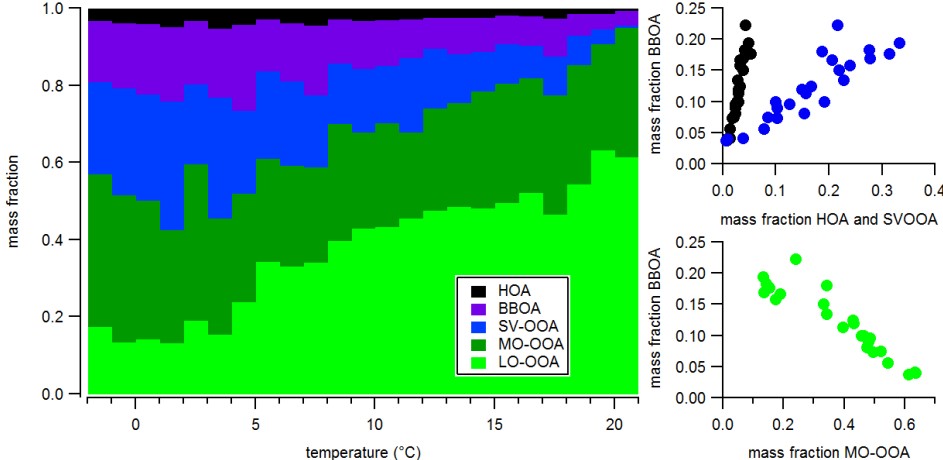

**Figure 6: Temperature dependency of the identified factors' contribution to OA (left) and their individual correlations (right). The OA mass concentrations were averaging to a one-degree Celsius resolution.**


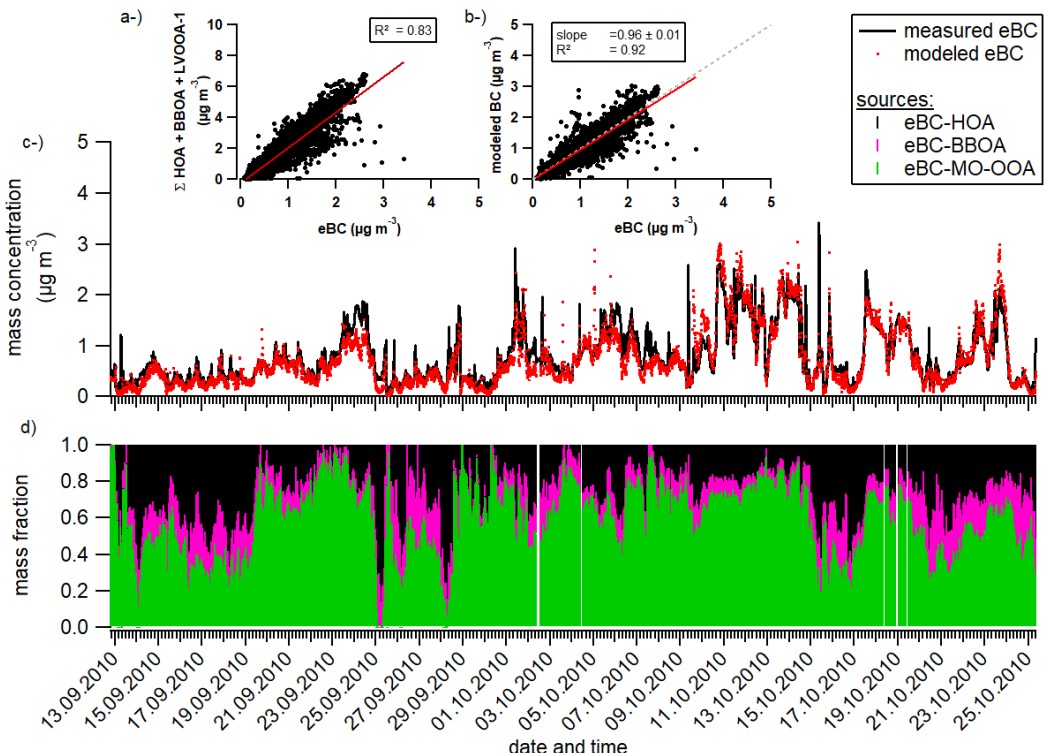

**Figure 7:** Contribution of the different organic factors to the eBC mass concentration. The scatter plots present the correlation between the sum of the OA factors and the measured eBC (a), and the estimated eBC concentration compared to the measured one (b). Time series show comparisons between measured and modeled eBC (c) and contributions of each source to the modeled eBC concentration (d). The correlation curves (red lines) were calculated using the least orthogonal distance fit method.

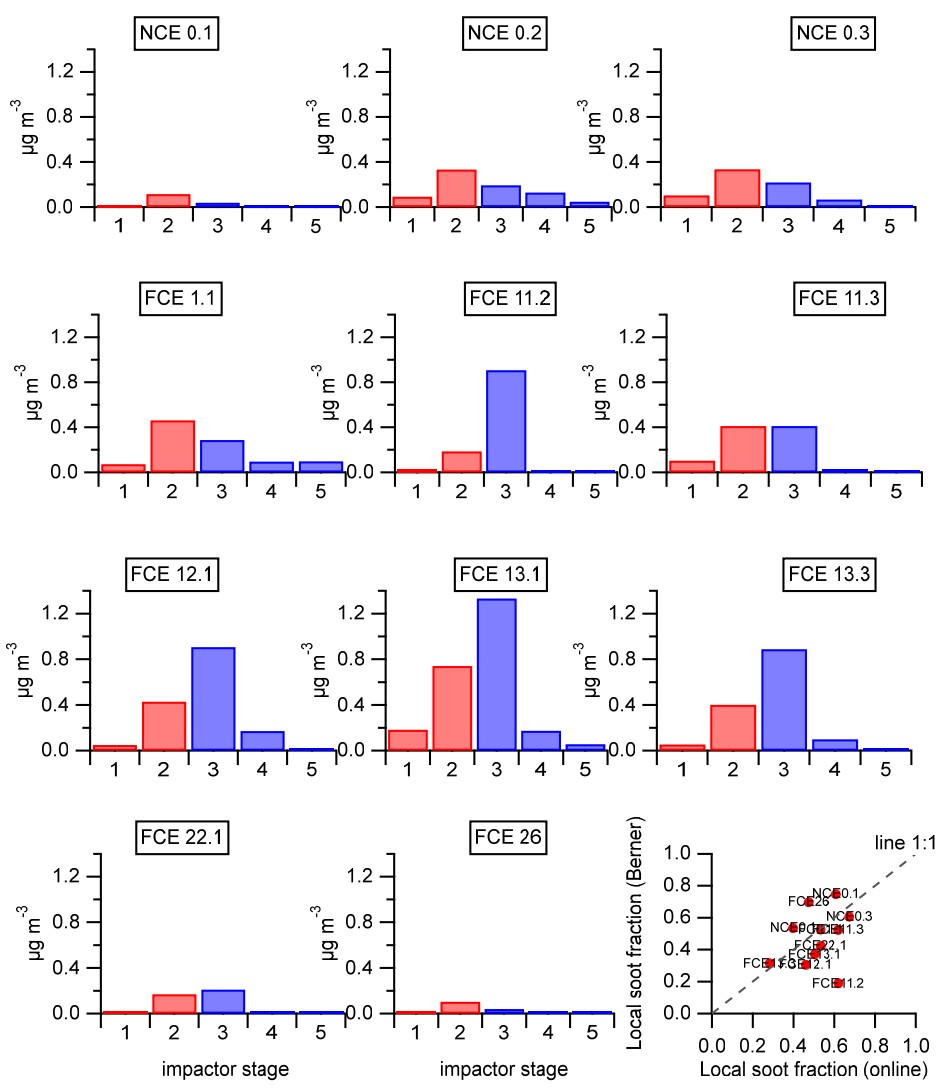

**Figure 8: Overview of the EC size distribution measured by the 5-stages Berner impactor. Color corresponds to the following EC classification: red = local and blue = regional/transport. The scatter plot on the bottom right shows the comparison between the local soot fractions estimated using the two different approaches: Berner impactor (y-axis) and on-line multilinear regression (x-axis).**




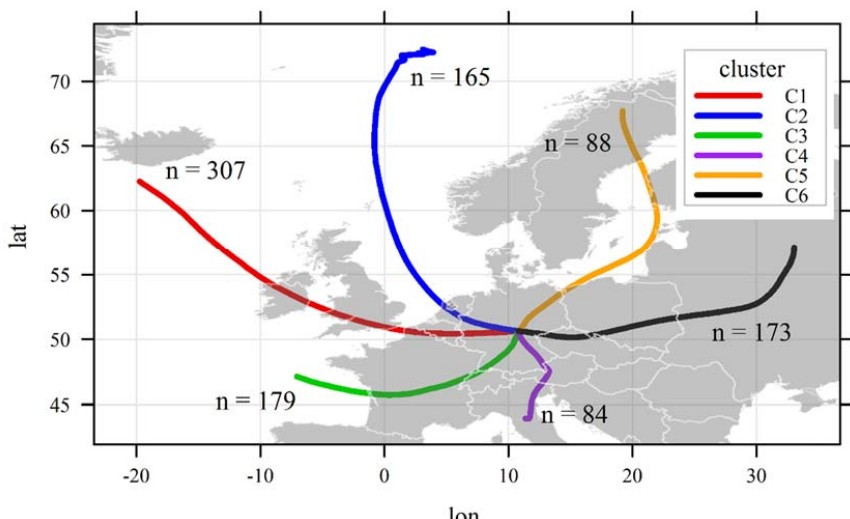


**Figure 9: Cluster results of the 96 h backward air mass trajectories calculated for the entire sampling period. The "n" indicates the sum of air mass trajectories associated with each cluster.**





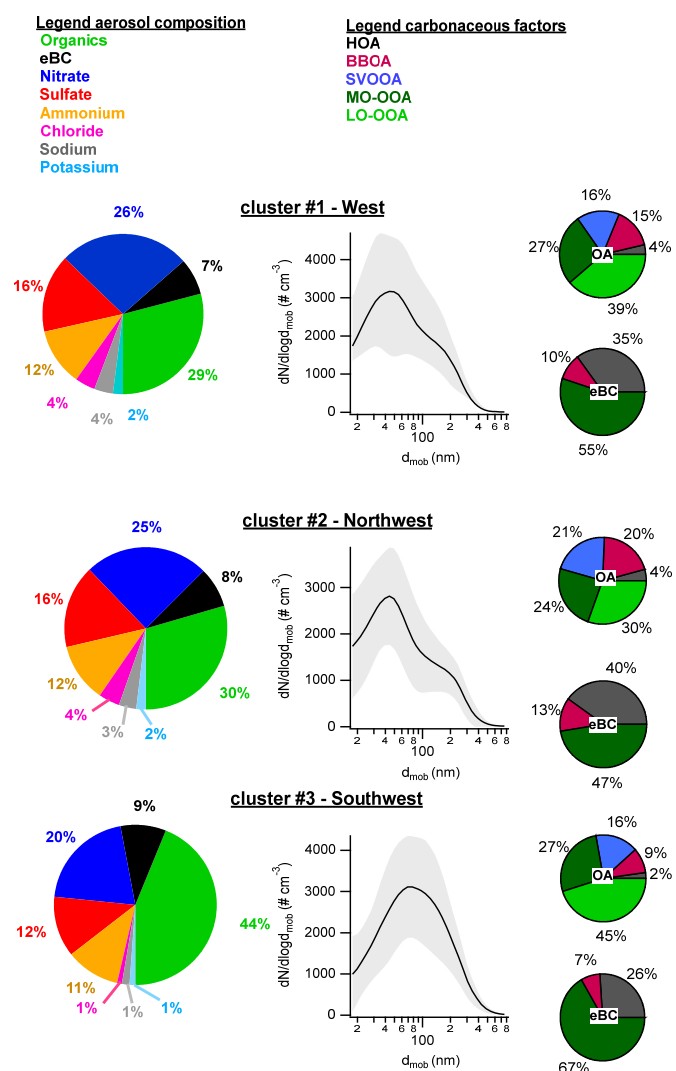

**Figure 10: Overview of the chemical composition and the PNSD for each cluster: on the left, the mean estimated PM₁₀ aerosol particle chemical composition; in the middle, the averaged particle number size distribution (± standard deviation in grey); and on the right, the source apportionment results for organic aerosol (top right) and eBC (bottom right).**






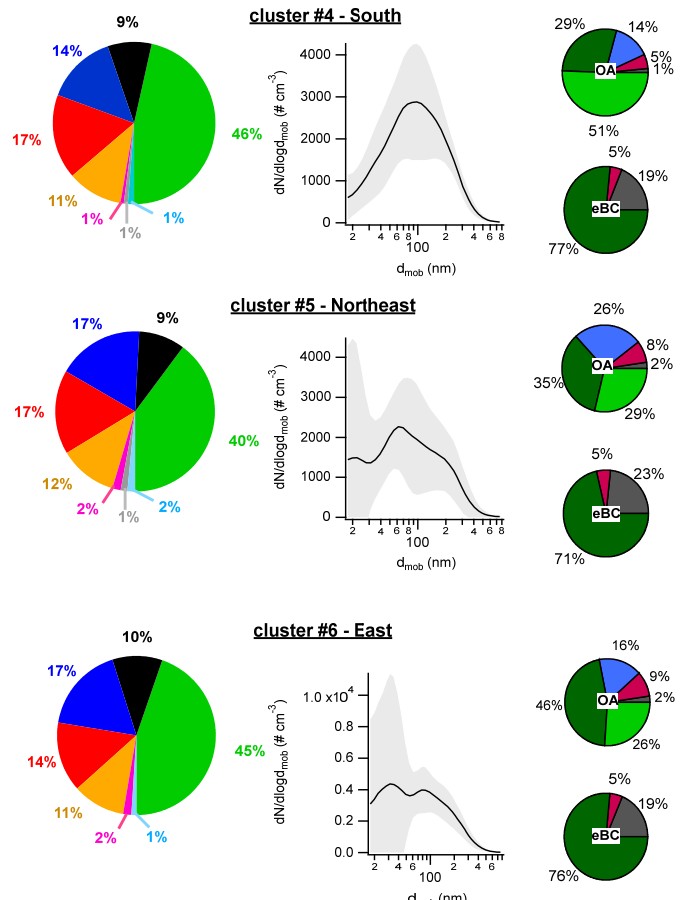


**Figure 10: Continued**