# Peer review of "Source apportionment and impact of long-range transport on carbonaceous aerosol particles in Central Germany during HCCT-2010"

_Atmospheric Chemistry and Physics, 2020_

## Referee Comment (RC1) · Anonymous Referee #1 · 18 Sep 2020

Poulain and coauthors describe a measurement campaign at a forest site in central Germany in 2010 involving the investigation of sources of carbonaceous aerosols using aerosol mass spectrometer (organic and inorganic aerosol) and multi angle absorption photometer (equivalent black carbon) measurements. Particle number-size distributions and light scattering were also measured using a dual mobility particle sizer and a nephelometer, respectively. Although originally chosen as an upwind site for a cloud processing study the dataset is also well suited to source apportionment of local and transported carbonaceous aerosols, which is the focus of this work. Good

mass closure was obtained for AMS+MAAP data relative to expected mass concentrations derived using size distribution data and composition-dependent density values. AMS mass concentrations were observed to agree well with supporting on-line measurements (Monitor for AeRosol and Gases in Ambient Air, MARGA) and off-line size-resolved filter analysis (ion chromatography/UV-Vis and OC/EC analysis). Organic aerosol as measured by the AMS was apportioned using PMF/ME-2 constrained with reference mass spectra for hydrocarbon-like organic aerosol (HOA) and biomass burning organic aerosol (BBOA). Five factors were derived: HOA, BBOA and three oxidized organic aerosol factors: semivolatile (SV-OOA), less oxidized (LO-OOA) and more oxidized (MO-OOA). eBC was found to be predominantly associated with long-range transport through multiple linear regression analysis, which is somewhat unexpected, and this observation is supported by the size-dependence of eBC during different air mass periods. Under marine air masses, locally emitted carbonaceous aerosol sources become more important in terms of fractional composition, however continental air masses from the East result in the worst local air quality at the site. Overall, I find the manuscript to be well written and the quantification and apportionment procedures are rigorous and comprehensive. The dependence of aerosol composition on air mass origin is established well and the findings reported here are a useful reference point for central European background sites. I only have a few minor comments.

How do the size distribution shapes for the impactor off-line analysis (Figure 2) agree with the AMS size distributions for nitrate/sulfate/ammonium/OC(or OA)?

Figure 2 caption define NCE and FCE here also

Figure 2: different colour for sodium needed here

Replace size cutting with size cut-off throughout the text

Line 230: The temperature dependence could also be related to less evaporation or oxidation of locally produced vehicle exhaust HOA. Is there more local wind stagnation at lower temperatures? This could also boost the contribution of local vehicle emissions.

Figure 8: add the regression statistics
* * *

---

## Referee Comment (RC2) · Anonymous Referee #2 · 15 Oct 2020

This manuscript presents a thorough analysis of the aerosol composition and their source apportionment at a mountain forest site in Central Germany using detailed aerosol and gas measurements for 40+ days in late-2010. The topic is important, the methodology is clear, and the findings are very well presented. By using detailed particle composition, organic aerosol source apportionment (ME2), and back trajectory analysis, Poulain et al. provide insights into sources of aerosol at this site. Among other things, their findings on more than half of Equivalent black carbon (eBC) coming from long-range transport is especially interesting and potentially relevant for ongoing and

future studies as well. I think the importance and quality of this manuscript warrants its publication in Atmospheric Chemistry and Physics.

I only have some minor comments:

Fig 1: Consider including Boundary Layer Height (BLH) timeseries either in Fig1 or in Fig S5. Reanalysis BLH (can easily be obtained from ECMWF's ERA5) seems to suggest potential role of changes in BLH height on the total aerosol mass loadings for the observed period.

Mention measurement period in *Introduction* or *Site and instrumentations*.

FigS5: Subplot-4 check colors.

---

## Author Response (AR1)

Answers to reviewer #1

Poulain and coauthors describe a measurement campaign at a forest site in central Germany in 2010 involving the investigation of sources of carbonaceous aerosols using aerosol mass spectrometer (organic and inorganic aerosol) and multi angle absorption photometer (equivalent black carbon) measurements. Particle number-size distributions and light scattering were also measured using a dual mobility particle sizer and a nephelometer, respectively. Although originally chosen as an upwind site for a cloud processing study the dataset is also well suited to source apportionment of local and transported carbonaceous aerosols, which is the focus of this work. Good mass closure was obtained for AMS+MAAP data relative to expected mass concentrations derived using size distribution data and composition-dependent density values. AMS mass concentrations were observed to agree well with supporting on-line measurements (Monitor for AeRosol and Gases in Ambient Air, MARGA) and off-line size-resolved filter analysis (ion chromatography/UV-Vis and OC/EC analysis). Organic aerosol as measured by the AMS was apportioned using PMF/ME-2 constrained with reference mass spectra for hydrocarbon-like organic aerosol (HOA) and biomass burning organic aerosol (BBOA). Five factors were derived: HOA, BBOA and three oxidized organic aerosol factors: semivolatile (SV-OOA), less oxidized (LO-OOA) and more oxidized (MO-OOA). eBC was found to be predominantly associated with longrange transport through multiple linear regression analysis, which is somewhat unexpected, and this observation is supported by the size-dependence of eBC during different air mass periods. Under marine air masses, locally emitted carbonaceous aerosol sources become more important in terms of fractional composition, however continental air masses from the East result in the worst local air quality at the site. Overall, I find the manuscript to be well written and the quantification and apportionment procedures are rigorous and comprehensive. The dependence of aerosol composition on air mass origin is established well and the findings reported here are a useful reference point for central European background sites.
I only have a few minor comments.

We would like to thank the referee for his/her constructive comments and suggestions made to improve and clarify our manuscript. Our responses are given below. For clarity, comments from the referee are in black, our responses in blue, and change on the text of the manuscript in **bold blue**.

R1: How do the size distribution shapes for the impactor off-line analysis (Figure 2) agree with the AMS size distributions for nitrate/sulfate/ammonium/OC(or OA)?
A1: Comparison between AMS and Berner impactor size distribution was added to the supplementary information for each FCE and NCE as followed:

[Figure]

[Figure]

[Figure]

**Figure SI-5:** Comparison between the AMS and Berner impactor size distribution for organic (organic mass for the AMS and OC for the Berner impactor), nitrate, sulfate, and ammonium for the different FCE and NCE events.

The following text discussing the comparison was included in the supplementary information section SI-3:

The MARGA ($PM_{10}$, Fig. SI-2 & SI-3) and Berner impactor $PM_{1.2}$ (sum of the three first stages, Fig. SI-4) mass concentration of ammonium, nitrate, and sulfate also present an excellent correlation with the AMS measurements regarding individual instrumental limitations. This includes the limited number of samples and the reduced sampling time of the Berner impactor, as well as the specific upper size cut-off (near-$PM_1$ for the AMS, $PM_{1.2}$ for the Berner impactor, and $PM_{10}$ for MARGA). **The size distribution measured by the Berner-impactor and the AMS was compared to each other for organic (Organic mass for the AMS, OC for the Berner impactor), nitrate, sulfate, and ammonium (Fig. SI-5). Similar size distributions were obtained for the inorganic species, except for FCE 12.1 which shows higher nitrate and ammonium mass concentrations on stage 3 compared to the AMS size distribution. A**

comparison between the two organics measurement methods is more difficult since the AMS directly measures the organic mass concentration, while the OC from the Berner impactor was obtained according to the two-step thermographic method (VDI standard). Moreover, the presence of different nitrate salts can also explain the discrepancy between AMS and MARGA (see discussion on main text).

R2:Figure 2 caption define NCE and FCE here also Figure 2: different colour for sodium needed here
A2: The definition of NCE and FCE was added to the figure caption and the color code of the different chemical species (including sodium) was changed.

[Figure]

**Figure 2: Size distribution of OC, EC, and major water-soluble ions from Berner impactor measurements for the different full cloud events (FCE) and non cloud event (NCE)**

R3: Replace size cutting with size cut-off throughout the text
A3: We replaced it in the main text, figures, and supplementary information.

R4 Line 230: The temperature dependence could also be related to less evaporation or oxidation of locally produced vehicle exhaust HOA. Is there more local wind stagnation at lower temperatures? This could also boost the contribution of local vehicle emissions.

A4: Thanks for pointing out this aspect. This is true that during low temperature periods, slower evaporation of the vehicle exhaust, as well as their oxidation, might be expected. Moreover, during the cold period, the wind speed was most of the time below 1 m s$^{-1}$, which could also contribute to lead higher local emission mass concentration by reducing the dilution effect. Moreover, during the entire campaign, relatively low wind speed was reported ranging up to 5 m s$^{-1}$.

The text was modified as follows:
This temperature dependency indicates that HOA should be mostly associated with local residential house heating rather than **a significant increase in car emissions. Moreover, the low temperature period can also lead to an artificial increase of the HOA concentration by slowing down the evaporation process of the emitted particles as well as their oxidation processes. Finally, the cold period was also associated with low wind speed and stable stratification (Fig. 1) (Tilgner et al., 2014), which might also contribute to higher concentrations of locally emitted aerosol by reducing both mixing and transport processes.**

R5: Figure 8: add the regression statistics
A5: regression slope and statistics are now included in the figure. The figure caption was modified accordingly.

[Figure]

Figure 8: Overview of the EC size distribution measured by the 5-stages Berner impactor. Color corresponds to the following EC classification: red = local and blue = regional/transport. The scatter plot on the bottom right shows the comparison between the local soot fractions estimated using the two different approaches: Berner impactor (y-axis) and on-line multilinear regression (x-axis). **Regression (black line) was made using the least orthogonal distance fit method.**

Reference

Tilgner, A., Schone, L., Brauer, P., van Pinxteren, D., Hoffmann, E., Spindler, G., Styler, S. A., Mertes, S., Birmili, W., Otto, R., Merkel, M., Weinhold, K., Wiedensohler, A., Deneke, H., Schrodner, R., Wolke, R., Schneider, J., Haunold, W., Engel, A., Weber, A., and Herrmann, H.: Comprehensive assessment of meteorological conditions and airflow connectivity during HCCT-2010, Atmos. Chem. Phys., 14, 9105-9128, 10.5194/acp-14-9105-2014, 2014.

Answers to reviewer #2

This manuscript presents a thorough analysis of the aerosol composition and their source apportionment at a mountain forest site in Central Germany using detailed aerosol and gas measurements for 40+ days in late-2010. The topic is important, the methodology is clear, and the findings are very well presented. By using detailed particle composition, organic aerosol source apportionment (ME2), and back trajectory analysis, Poulain et al. provide insights into sources of aerosol at this site. Among other things, their findings on more than half of Equivalent black carbon (eBC) coming from long-range transport is especially interesting and potentially relevant for ongoing and future studies as well. I think the importance and quality of this manuscript warrants its publication in Atmospheric Chemistry and Physics.

We would like to thank the referee for his/her constructive comments and suggestions made to improve and clarify our manuscript. Our responses are given below. For clarity, comments from the referee are in black, our responses in blue, and change on the text of the manuscript in **bold blue**.

I only have some minor comments:
R1: Fig 1: Consider including Boundary Layer Height (BLH) timeseries either in Fig1 or in Fig S5. Reanalysis BLH (can easily be obtained from ECMWF's ERA5) seems to suggest potential role of changes in BLH height on the total aerosol mass loadings for the observed period.
A1: As suggested, the boundary layer height (BLH) time series was included in Fig. 1. Here the BLH time series was retrieved from the HYSPLIT GDAS (1 degree resolution) input, which was used for the trajectory analysis.

The text was modified as follow:
Section: 2.3 Back-trajectories and cluster calculations
The 96 h back trajectories were used to determine the influence of the air mass origin on aerosol. The trajectories were calculated for every hour from 13 September until 24 October 2010 for the altitude of 500 m above model ground with the NOAA Hybrid Single Particle Lagrangian Integrated Trajectory (HYSPLIT-4) Model (http://www.ready.noaa.gov/ready/hysplit4.html; Draxler and Hess, 2004) using the 1 degree resolution GDAS input data. The different back-trajectory clusters were calculated using the program R (http://www.r-project.org/; R Core Team, 2013) with the package openair (http://www.openair-project.org; Carslaw and Ropkins, 2012;Ropkins and Carslaw, 2012). **The same GDAS input data was used to retrieve the boundary layer height (BLH) at the sampling site from the HYSPLIT model output.**

Section: 3.1.1 Overall AMS-MAAP time series
Aerosol particle chemical composition (mass concentration and mass fraction) as measured by AMS and MAAP as well as the particle number size distribution over the entire time-period are shown in Figure 1. On average, the near-PM$_1$ particulate chemical composition was principally made-up of organic aerosol, OA (41 % of the total mass, Fig. 1). Sulfate and nitrate have quite similar contributions (19 % and 18 %, respectively). The rest of the aerosol particle mass concentration was made of ammonium (11 %), eBC (10.0 %), and chloride (1 %). Despite their similar contribution to the particle mass fraction, sulfate and nitrate showed a clear time dependency (Fig. 1). Although sulfate dominates the inorganic fraction at the beginning of the measurement period, nitrate becomes more important over time. This can be directly linked to a decrease of temperature during the sampling period (Fig. SI-5), inducing a change in nitrate partitioning between gas and particle phase. A last factor that must be considered is the decrease of solar radiation from summer to winter, influencing the photochemical formation of sulfate. **Variation of the BLH over the sampling period can also influence the local PM mass concentration (Fig.1). At the beginning of the campaign, the BLH reached above 1000 m during daytime, while the maximum altitude of the BLH decreased to below 800 m later on. This decrease**

**in the maximum altitude of the BLH certainly contributes to the observed increase of the overall PM mass concentration during the day by reducing the ventilation effect. However, it is important to note that high uncertainties on the absolute value of the BLH for such a mountain area have to be expected due to the 1 degree resolution of the GDAS input data.** Variations of the organics and eBC mass concentration over the sampling period will be discussed in sections 3.2 and 3.3, respectively.

[Figure]

Figure 1: Time series of the ambient temperature (a), **estimated boundary layer height (BLH) obtained from HYSPLIT GDAS input (b),** the particulate near-PM$_1$ chemical composition as measured by the AMS and completed by MAAP for equivalent black carbon (c), the corresponding mass fraction (d), and particle number size distribution (e) during HCCT-2010 at the site of Goldlauter. The colored bars and numbers at the top refer to the six different air mass clusters (see section 3.4), and the grey bars refer to the different cloud and non-cloud events as defined in Table SI-2.

R2: Mention measurement period in Introduction or Site and instrumentations.
A2: The following sentence was modified in the introduction section to include reviewer's suggestion:
The measurements **took place on September-October 2010 as part** of the "Hill Cap Cloud Thuringia 2010" (HCCT-2010) experiment, which aimed to investigate the impact of cloud processing to aerosol physico-chemical properties.

R3: FigS5: Subplot-4 check colors.
A3: Colors for global radiation and relative humidity were corrected in Fig. S4.

[Figure]

Figure SI-5: Overview of the meteorological conditions during the sampling period.

[revised manuscript text omitted]

**SI-1 Instrumental set-up and analytical methods**

**Table SI-1: Instrumentation used for trace gas measurements.**

| parameter | Type | Time resolution | Detection limits | Technique/Method/size cutt-off | Measured species |
|---|---|---|---|---|---|
| Gas samplers [*] | | | | | |
| Ozone | TE49C-TL (Thermo Fischer Scientific Inc.) | < 2 min | 1 ppb | UV-absorption | Ozone |
| $NO_x$ (NO/$NO_2$) | TE42S Thermo (Fischer Scientific Inc.) | 0.5 min | 0.5 ppb | Chemiluminescence | NO and $NO_2$ |
| $SO_2$ | TE43C-TL (Thermo Fischer Scientific Inc.) | <2 min | 0.2 ppb | UV-fluorescence | $SO_2$ |
| CO | ML 9830 (Monitor Europe) | | | | |
| Water soluble gases | MARGA | 1 hour | | Denuder sampling for gases and online IC analysis | $NH_3$, HCl, HONO, $HNO_3$, $SO_2$ |
| VOC | | 2 h during IOPs | | Thermal desorption GC-FID | 26 NMHC (C2-C8) |

[*] $SO_2$ and $NO_x$ analyzers were calibrated using test gas cylinders (air liquid, Germany), $NO_2$ by a gas-phase-titration system (Sycos K/GPT; Ansyco GmbH, Germany), and an $O_3$ analyzer by the calibrator system TE49PS

**SI-2 Sample preparation and chemical analysis of the 5-stage Berner-impactor**

Samples of the 5-stage Berner-impactor were collected on aluminum foils (Table SI-2). Each foil was weighed after a 72 h equilibration under constant temperature (20±1 °C) and humidity (50±5 %) before and after collection using an electronic microbalance (UMT 2, Mettler Toledo, Switzerland) with a reading precision of 0.1 µg and a reproducibility of 1 µg. A part of the impactor foils was extracted with 1.5 ml MilliQ-water (> 18.2 MΩ cm; 15 min shaker, 15 min ultrasonic bath, 15 min

30 shaker). Sample extracts were then filtered through a 0.45 µm disposable syringe filter to remove insoluble materials prior to ion analysis by ion chromatography (ICS300, Dionex, USA) for cations (Column CS16, eluent methane sulfonic acid) and anions (Column AS18, eluent KOH). Calibrations were carried out daily using a four point's standard diluted from a stock solution (Fluka, Switzerland). The detection limits for all ions measured by conductivity detection were within 0.1 mg l$^{-1}$, except for sulfate and nitrate (0.2 mg l$^{-1}$). Nitrite was detected using UV/VIS detection (VWD-1, Dionex) with a detection

35 limit of 0.1 mg l$^{-1}$, which leads to a general detection limit ranging between 0.005 and 0.05 µg m$^{-3}$ depending on the sampling volume and species. Blank corrections were made according to the analyzed field blank impactor foils. The organic carbon (OC) and elemental carbon (EC), in sum total carbon (TC), analyses were made using a carbon analyzer type C-mat 5500 with a non-disperse infrared detector (NDIR) (Ströhlein, Germany) based on a modification of the German VDI guideline 2465 (Gnauk et al., 2008). Finally, sugars and anhydrosaccharides (e.g. levoglucosan, galactosan and mannosan) were analyzed

40 from the water extract prepared for ion chromatography using an ICS3000 system (Dionex, U.S.A.) equipped with a pulsed amperometric detector (Engling et al., 2006;Iinuma et al., 2009).

**Table SI-2: Berner-impactor measurements periods associated to full cloud events (FCE) and non-cloud events (NCE) taken during the sampling periods. More details on the FCEs and NCEs can be found in Tilgner et al. (2014).**

|  | FCE 1.1 | NCE 0.1 | NCE 0.2 | FCE 11.2 | FCE 11.3 | FCE 12.1 | FCE 13.1 | FCE 13.3 | FCE 22.1 | NCE 0.3 | FCE 26 |
|---|---|---|---|---|---|---|---|---|---|---|---|
| Starting time | 14.09.2010 11:00 | 15.09.2010 18:00 | 20.09.2010 11:25 | 01.10.2010 22:30 | 02.10.2010 14:30 | 05.10.2010 08:30 | 05.10.2010 19:15 | 06.10.2010 12:15 | 19.10.2010 21:30 | 21.10.2010 14:15 | 24.10.2010 01:30 |
| Stopping time | 15.09.2010 02:00 | 15.09.2010 23:30 | 20.09.2010 20:30 | 02.10.2010 05:30 | 02.10.2010 19:30 | 05.10.2010 13:00 | 06.10.2010 04:30 | 07.10.2010 03:15 | 20.10.2010 03:30 | 21.10.2010 22:15 | 24.10.2010 11:45 |
| Back-trajectory Cluster | C1 | C1 | C1 | C3 | C3 | C4 to C3 | C3 | C3 to C4 | C1 | C1 | C3 to C2 via C1 |

**SI-3 AMS data validation**

Prior to mass closure analysis, conversion of the particle number concentration of the T-MPSS to the volume concentration was made assuming spherical particles, and to the mass concentration using a time dependent density estimated using the

50 equation of Salcedo et al. (2006) and based on the measured $PM_1$ chemical composition as previously described in Poulain et al. (2014). A good correlation was obtained (slope of 0.93, $R^2 = 0.94$, Fig. SI-1), indicating that non-detected compounds (i.e. AMS refractory compounds except eBC) do not significantly contribute to the $PM_1$ mass concentration.

[Figure]

55 **Figure SI-1: Mass closure between online aerosol chemical composition (AMS and MAAP) and TDMPS estimated mass concentration (bottom and insert). Time and chemical dependent density (top) was used to convert the volume concentration of the TDMPS into mass concentration. The correlation curve (black line) was calculated using the least orthogonal distance fit method**

The MARGA ($PM_{10}$, Fig. SI-2 & SI-3) and  Berner-impactor $PM_{1.2}$ (sum of the three first stages, Fig. SI-4) mass

60 concentration of ammonium, nitrate, and sulfate also present an excellent correlation with the AMS measurements regarding individual instrumental limitations. This includes the limited number of samples and the reduced sampling time of the  Berner-impactor, as well as the specific upper size  cut-off (near-$PM_1$ for the AMS, $PM_{1.2}$ for the  Bernerimpactor, and PM$_{10}$ for MARGA). The size distribution measured by the Berner-impactor and the AMS was compared to each other for organic (Organic mass for the AMS, OC for the Berner-impactor), nitrate, sulfate, and ammonium (Fig. SI-5). Similar size distributions were obtained for the inorganic species, except for FCE 12.1 which shows higher nitrate and ammonium mass concentrations on stage 3 compared to the AMS size distribution. A comparison between the two organics measurement methods is more difficult since the AMS directly measures the organic mass concentration, while the OC from the Berner-impactor was obtained according to the two-step thermographic method (VDI standard). Moreover, the presence of different nitrate salts can also explain the discrepancy between AMS and MARGA (see discussion on main text).

[Figure]

**Figure SI-2: Sulfate (top) and ammonium (bottom) time series measured by AMS (colored line) and MARGA (black line).**

[Figure]

75 **Figure SI-3: Scatter plots of the nitrate, sulfate, and ammonium mass concentrations measured by AMS and MARGA. The correlation curves (red lines) were calculated using the least orthogonal distance fit method.**

[Figure]

80 **Figure SI-4: Comparison of AMS and impactor measurements (stages 1-3, i.e. PM1.2) during the different cloud and non-cloud events (see text for details on sampling periods). The correlation curves (red lines) were calculated using the least orthogonal distance fit method**

85

[Figure]

Figure SI-5: Comparison between the AMS and Berner-impactor size distribution for organic (organic mass for the AMS and OC for the Berner-impactor), nitrate, sulfate, and ammonium for the different FCE and NCE events.

[Figure]

95

**Figure SI-5: Continued**

[Figure]

**Figure SI-5: Continued**

**SI-4 Overview of the meteorological conditions**

105

[Figure]

**Figure SI-65: Overview of the meteorological conditions during the sampling period.**

110

**SI-5 Organic aerosol source apportionment**

Source apportionment was performed on the high-resolution organic mass spectra dataset using the Multi-linear Engine (ME-2) model developed by Paatero (1999) and using the Source Finder tool Sofi4.9 (Canonaco et al., 2013) developed at the Paul Scherrer Institute (Switzerland). The source apportionment was made following the recommendation of Crippa et al. (2014):

115 for the first time a non-constrained model approach was investigated and since primary factors were not properly resolved in this first model, a partially constrained approach was investigated during a second time.

**SI-5-1 Non-constrained model (PMF)**

In the non-constrained ME-2 model, solutions were investigated in a range from 2 to 10 factors, each within 20 seeds (Fig. SI-

120 67). The best solution was obtained for the 4-factors solution and the different factors were identified as primary OA (POA; 20 % of the total OA), semi-volatile OA (SV-OOA; 17 %), and 2 OOAs (named LV-OOA1 and LV-OOA2; 28 % and 34 %, respectively) (Fig. SI-78). The POA factor was identified based on its low oxidation (O:C = 0.27) and presence of typical tracers from two primary OAs (Hydrocarbon-like OA (HOA) with the m/z 55 and 57 and Biomass Burning OA (BBOA) with m/z 60 and 73). Moreover, similarity between its diurnal profile and the ones of the different anthropogenic emission tracers

125 (eBC, CO, and NO$_2$, Fig. SI-89) confirm its primary origin. The presence of the BBOA tracers (m/z 60 and m/z 73) on the POA mass spectra indicates a dominant biomass burning influence on the primary OA sources. Increasing the number of factors to 5 leads to an additional split of the POA factor without providing clear HOA and BBOA factors. However, it is interesting to note the relative stability of the two OOA mass spectra in the range from 3- to 5-factors (Fig. SI-910). Changes on their individual concentrations highlight the influence of the additional splitting when increasing the number of factor

130 solutions. This stability indicates that these two factors can easily be extracted from the OA matrix by the model and therefore, their identification can be considered as quite robust in the range of 3- to 5-factors solutions. By increasing the factor's number above 5, additional splits of the two OOAs were observed without providing either with more details on their sources or a better identification of the POAs.

The LV-OOA1 factor correlates better with eBC than the POA factor (R² = 0.80 for LV-OOA1 vs. eBC, while R²=0.38 for

135 POA vs. eBC). Since the LV-OOA1 is quite oxygenated (O:C = 0.91) and does not show a similar diurnal pattern to eBC (Fig. SI-78 & SI-89), it is quite difficult to link this factor to any local primary emission. Consequently, and due to its stability over the investigated factor solutions, it should be related to process anthropogenic and long-range transport of polluted air masses (see discussion in section 3.2.4 of the manuscript). Therefore, it clearly indicates the presence of two distinct sources of eBC: a local one related to the POA factor and a second one related to regional or transported aged primary emissions associated to

140 LV-OOA1 (details on the eBC source apportionment can be found in the dedicated section of the manuscript).

[Figure]

**Figure SI-76:** **Variation of the factor contributions (top) and Q/Q_exp (bottom) over the investigated solution area for the unconstrained model.**

[Figure]

**Figure SI-7: Time series of the PMF 4-factor solutions and comparison with collocated measurements (top). The corresponding high-resolution mass spectra colored by fragment family codes is presented on the bottom.**

[Figure]

**Figure SI-98: Diurnal pattern of the identify factors and collocated measurements for the non-constrained model.**

[Figure]

[Figure]

**Figure SI-9: Comparison of the time series (top) and mass spectra (bottom in relative intensity) of the two LV-OOA factors ranging from 3- to 6-factor solutions.**

**SI-5-2 Partly constrained model**

**SI-5-2-1 Constraining HOA**

Two different POA factors were expected based on the local emissions: an HOA factor related to traffic and fossil fuel combustion and biomass burning OA (BBOA). The presence of biomass burning OA is also confirmed when looking at the contribution of the fragment 60 to the total OA ($f_{60}$), which went above the threshold value of 0.3 % suggested by Cubison et al. (2011) (not shown). In order to better distinguish the different primary sources, the source apportionment model was therefore partly constrained, using as reference the HOA mass spectra from Mohr et al. (2012) obtained at Barcelona (Spain) and available on the reference mass spectra database (http://cires.colorado.edu/jimenez-group/HRAMSsd/, Ulbrich et al., 2009). Results were investigated using a factor number ranging from 2 to 9 with an anchor for the HOA factor ranging from 0.05 to 0.5 (Fig. SI-11 & SI-12). Contribution of HOA to total OA was relatively stable over the entire investigated factor solution as well as over the different a-values, as can be seen in Figure SI-12. Therefore, in the following, a solution with an a-value of 0.1 was considered. This corresponds to a quite constrained mass spectrum. From the 5-factors solution a possible BBOA factor can be suspected based on the high contribution of m/z 60 and 73. However, comparing the mass spectra of the 5-factor solutions BBOA with reference ones (Fig. SI-13) shows a very high contribution of $CO_2^+$ fragments, which might suggest a possible contribution of OOA and is therefore a non-properly resolved factor.

Additionally, the three OOAs (SV-OOA, LV-OOA1, and LV-OOA2) already identified in the non-constrained model were also found. The two LV-OOAs are extremely close to the ones identified in the non-constrained analysis (Fig. SI-12; LV-OOA1: $R^2$= 0.97, slope 1.05; and LV-OOA2 $R^2$= 0.95, slope 0.98 for their respective time series comparison). A larger discrepancy was observed for SV-OOA when comparing it to the non-constrained model ($R^2$= 0.73, slope 0.67). This could result from a possible contribution of the BBOA factor to the non-constrained SV-OOA factor, confirming the non-ideal apportionment of the BBOA factor.

Increasing the number of factors leads mainly to an additional split of the OOAs. Interestingly, a well-defined BBOA mass spectrum was found for the 9-factors solution. This BBOA factor is in agreement with the reference mass spectra proposed by Ng et al. (2011), as well as with the averaged BBOA factor obtained during the EUCAARI project in Europe Crippa et al. (2014) (Fig. SI-13). Consequently, a third model was built, constraining the HOA mass spectra as previously, as well as a BBOA factor using the identified BBOA mass spectra on the 9-factors solution. Since the BBOA directly results from the present dataset, the BBOA mass spectrum was constrained with a fixed a-value of 0.1.

[Figure]

190    **Figure SI-11: Variation of the Q/Q~exp~ (top) and factor contributions (bottom) over the investigated solution area with constrained HOA.**

[Figure]

195  **Figure SI-11: Comparison of the time series (top, in µg m⁻³) and mass spectra (bottom) of the 5-factors solution partly-constrained approach (fixed HOA with a-value = 0.1) and the non-constrained one (red).**

[Figure]

**Figure SI-12: Comparison of the identified BBOA factor for the 9-factors solution with reference mass spectra from Ng et al. (2011) and Crippa et al. (2014). The insert panel represents the scatter plots between the identified BBOA factor (after averaging to unit mass resolution (UMR)) and the 2 references.**

**SI-5-2-2 Constraining HOA and BBOA**

The investigation of the third model run is presented in Figure SI-14. Here, the 5-factors solution was retained as a final source apportionment result, which corresponds to the identification of the expected two primary OA (HOA and BBOA) and three additional OOA factors (SV-OOA, LO-OOA, and MO-OOA) (Fig. 5 on the manuscript). Both mass spectra and time series of the three OOA factors identified in this third model are similar to the ones obtained in the non-constrained model (Fig. SI-15 & 16). This confirmed the robustness of the OOAs identification, as well as the correct speciation of the POA factor identity on the non-constrained model. The larger variation in the elemental composition was found for the SV-OOA factor, which could be related to a small contribution from BBOA on the non-constrained results, as was mentioned before. The identification of the BBOA factor is confirmed by the comparison of the factor mass concentrations with the levoglucosan concentrations obtained from the off-line filter measurements made during IOPs (Fig. SI-17).

[Figure]

|215 **Figure SI-14̶1̶3̶: Variation of the Q/Q_exp (top) and factor contributions (bottom) over the investigated solution area with constrained HOA and BBOA.**

[Figure]

220

**Figure SI-1514:** Comparison of 5-factors solution time series with constrained HOA and BBOA (left axis) with the previously identified ones (right axis) obtained during the 9-factors solution (with fixed HOA) for HOA and BBOA (red) and the PMF results (non-constrained) for the two OOAs (green).

[Figure]

225 **Figure SI-16: Comparison of the OOAs mass spectra from the final HOA and BBOA constrained factor solutions with the ones previously obtained on the non-constrained solution.**

[Figure]

**Figure SI-16: Scatter plot of the identified factors (HOA, BBOA, and MO-OOA) vs. their corresponding tracers during intensive sampling periods. The correlation curves (red lines) were calculated using the least orthogonal distance fit method.**

230

[Figure]

**Figure SI-17: Continued**

[Figure]

235    **Figure SI-17: Diurnal profiles of the 5-factors solutions and their corresponding tracers.**

[Figure]

**Figure SI-1918: Scatter plot of the SV-OOA vs. anthropogenic NMHC gases. The correlation curves (red lines) were calculated using the least orthogonal distance fit method.**

240

245

**SI-6: Overview of the NMHC measurements**

**Table SI-3: Overview of the average gas-phase concentrations NMHC (in ppbv ± standard deviation) during full-cloud events (FCE) and non-cloud events (NCE).**

| | FCE 1.1 | NCE 0.1 | NCE 0.2 | FCE 11.2 | FCE 11.3 | FCE 12.1 | FCE 13.1 | FCE 13.3 | FCE 22.1 | NCE 0.3 | FCE 26 |
|---|---|---|---|---|---|---|---|---|---|---|---|
| Starting time | 14.09. 2010 12:44 | 15.09. 2010 18:00 | 20.09. 2010 11:25 | 01.10.2 010 22:33 | 02.10. 2010 14:33 | 05.10. 2010 08:30 | 05.10. 2010 19:15 | 06.10. 2010 12:15 | 19.10. 2010 21:30 | 21.10. 2010 14:15 | 24.10. 2010 01:30 |
| Stopping time | 15.09. 2010 00:00 | 15.09. 2010 23:30 | 20.09. 2010 20:30 | 02.10. 2010 05:30 | 02.10. 2010 20:00 | 05.10. 2010 13:00 | 06.10. 2010 04:30 | 07.10. 2010 02:45 | 20.10. 2010 03:30 | 21.10. 2010 22:15 | 24.10. 2010 11:45 |
| Cluster | C1 | C1 | C1 | C3 | C3 | C4 to C3 | C3 | C3 to C4 | C1 | C1 | C3 to C2 via C1 |
| **NMHC** | | | | | | | | | | | |
| Nb of samples | 5 | 3 | 6 | 4 | 3 | 2 | 5 | 0 | 3 | 3 | 6 |
| Acetylene | 0.61± 0.16 | 0.46± 0.13 | 0.62± 0.09 | 0.60± 0.03 | 0.87± 0.44 | 0.81± 0.04 | 0.82± 0.15 | | 0.71± 0.23 | 0.60±0. 13 | 0.55± 0.06 |
| Alkenes | 1.35± 1.15 | 0.79± 0.16 | 0.86± 0.11 | 1.36± 0.12 | 1.70± 0.58 | 1.77± 0.23 | 2.24± 0.54 | | 1.78± 0.37 | 1.72±0. 57 | 1.07± 0.44 |
| Aromatics | 0.60± 0.18 | 0.41± 0.08 | 0.44± 0.08 | 0.78± 0.11 | 0.88± 0.29 | 0.89± 0.08 | 1.09± 0.39 | | 0.64± 0.12 | 0.49± 0.07 | 0.31± 0.09 |
| $n$-alkanes | 3.36± 1.45 | 2.33± 0.66 | 2.64± 0.24 | 3.35± 0.34 | 2.97± 0.29 | 6.06± 0.28 | 5.02± 0.36 | No data | 3.35± 0.27 | 4.15± 0.37 | 2.96± 0.87 |
| $i$-alkanes | 0.66± 0.52 | 0.43± 0.19 | 0.36± 0.06 | 0.67± 0.07 | 0.49± 0.03 | 1.31± 0.09 | 1.09± 0.38 | | 0.61± 0.07 | 0.67± 0.06 | 0.37± 0.11 |
| Total NMHC | 6.59± 3.17 | 4.42± 1.03 | 4.91± 0.50 | 6.74± 0.59 | 6.90± 1.58 | 10.82± 0.72 | 10.25± 1.21 | | 7.08± 0.92 | 7.63± 0.81 | 5.26± 1.43 |
| Benzene-to-toluene ratio | 0.83 | 0.68 | 0.97 | 1.25 | 1.74 | 1.56 | 1.03 | | 1.26 | 1.26 | 1.20 |
| Xylene-to-toluene ratio | 0.58 | 0.35 | 0.47 | 0.74 | 0.70 | 0.64 | 0.58 | | 0.74 | 0.57 | 0.42 |

**SI-7 Influence of air mass origin on local aerosol and gas composition**

255 **Table SI-4: Gas phase tracer and particle mean composition for the different air mass clusters (all concentrations in μg m$^{-3}$)**

| cluster (hours) | C1 (307) | C2 (165) | C3 (179) | C4 (84) | C5 (88) | C6 (173) |
|---|---|---|---|---|---|---|
| region of air mass origin | West | Northwest | Southwest | South | Northeast | East |
| **Gas-phase measurements** | | | | | | |
| HCl mean (median) in μg m$^{-3}$ | 0.15 (0.11) | 0.09 (0.09) | 0.29 (0.21) | 0.36 (0.37) | 0.21 (0.21) | 0.13 (0.14) |
| Min. /Max. | 0.05 / 0.65 | 0.06 / 0.15 | 0.05 / 0.75 | 0.07 /0.62 | 0.21 /0.21 | 0.06 /0.30 |
| | | | | | | |
| HONO mean (median) in μg m$^{-3}$ | 0.76 (0.74) | 0.72 (0.67) | 0.67 (0.62) | 0.78 (0.76) | 0.38 (0.39) | 0.43 (0.32) |
| Min. /Max. | 0.11 / 2.44 | 0.10 / 2.42 | 0.17 / 1.81 | 0.16 /1.39 | 0.08 / 0.92 | 0.08 / 1.64 |
| | | | | | | |
| SO$_2$ mean (median) in μg m$^{-3}$ | 0.32 (0.24) | 0.30 (0.25) | 0.40 (0.26) | 0.37 (0.25) | 0.66 (0.21) | 2.48 (1.49) |
| Min. /Max. | 0.10 /1.46 | 0.10 / 0.68 | 0.1 / 2.00 | 0.11 / 1.71 | 0.10 / 3.50 | 0.14 / 13.27 |
| | | | | | | |
| HNO$_3$ mean (median) in μg m$^{-3}$ | 0.16 (0.13) | 0.14 (0.12) | 0.32 (0.23) | 0.25 (0.18) | 0.17 (0.11) | 0.34 (0.30) |
| Min. /Max. | 0.05 / 1.04 | 0.05 / 0.83 | 0.06 / 5.77 | 0.05 / 1.21 | 0.05 / 0.53 | 0.05 / 1.15 |
| | | | | | | |
| Ozone mean (median) in ppb | 42.69 (41.39) | 38.40 (40.27) | 39.96 (35.37) | 38.46 (33.13) | 42.77 (40.50) | 57.01 (60.01) |
| Min. /Max. | 8.73 / 81.09 | 7.86 / 71.6 | 7.77 / 94.78 | 8.28 / 99.75 | 14.92 / 81.13 | 8.01 / 90.73 |
| | | | | | | |
| NO mean (median) in ppb | 0.42 (0.15) | 0.32 (0.12) | 0.52 (0.28) | 0.95 (0.19) | 0.35 (0.21) | 0.71 (0.42) |
| Min. /Max. | 0.01 / 5.88 | 0.01 / 4.51 | 0.01 /8.08 | 0.01 / 7.78 | 0.01 / 1.47 | 0.01 / 4.88 |
| | | | | | | |
| NO$_2$ mean (median) in ppb | 5.17 (4.57) | 4.51 (4.19) | 5.87 (5.63) | 4.75 (4.64) | 4.53 (4.65) | 5.60 (5.21) |
| Min. /Max. | 0.96 / 16.04 | 1.04 / 13.19 | 1.86 / 13.79 | 1.76 / 10.17 | 1.05 / 11.07 | 1.88 / 15.22 |

**Table SI-4: Continued**

| cluster (hours) | C1 (307) | C2 (165) | C3 (179) | C4 (84) | C5 (88) | C6 (173) |
|---|---|---|---|---|---|---|
| region of air mass origin | West | Northwest | Southwest | South | Northeast | East |
| **Aerosol measurements in µg m$^{-3}$** | | | | | | |
| Total mass concentration mean (median) | 7.54 (7.61) | 6.38 (5.83) | 11.50 (9.93) | 11.38 (10.78) | 7.97 (5.75) | 12.65 (11.40) |
| Min. / Max. | 0.23 / 18.70 | 1.07 / 20.28 | 4.23 / 23.22 | 4.08 / 21.33 | 1.13 / 27.47 | 0.73 / 29.66 |
| | | | | | | |
| Organic (AMS) mean (median) | 2.07 (1.90) | 1.86 (1.57) | 4.99 (4.44) | 5.39 (5.05) | 3.18 (2.12) | 5.76 (5.66) |
| Min. /Max. | 0.18 / 5.06 | 0.28 / 6.62 | 1.52 / 12.0 | 2.30 / 11.36 | 0.30 / 9.73 | 1.19 /10.55 |
| | | | | | | |
| eBC (MAAP) mean (median) | 0.52 (0.49) | 0.50 (0.41) | 1.03 (0.91) | 1.01 (0.93) | 0.74 (0.41) | 1.31 (1.32) |
| Min /Max. | 0.08 / 1.5 | 0.10 / 1.83 | 0.31 / 2.22 | 0.56 / 3.07 | 0.11 / 2.31 | 0.16 / 3.2 |
| | | | | | | |
| Chloride (MARGA) mean (median) | 0.29 (0.21) | 0.27 (0.23) | 0.11 (0.10) | 0.08 (0.08) | 0.14 (0.12) | 0.21 (0.22) |
| Min. /Max. | 0.05 / 1.45 | 0.05 / 0.78 | 0.05 / 0.25 | 0.05 / 0.10 | 0.06 / 0.27 | 0.05 / 0.33 |
| | | | | | | |
| Nitrate (MARGA) mean (median) | 1.87 (1.67) | 1.55 (1.20) | 2.33 (1.92) | 1.64 (1.32) | 1.40 (0.88) | 2.25 (1.90) |
| Min. /Max. | 0.14 / 8.99 | 0.29 / 6.56 | 0.20 / 8.91 | 0.14 / 4.68 | 0.11 / 7.40 | 0.17 / 9.90 |
| | | | | | | |
| Sulfate (MARGA) mean (median) | 1.12 (1.11) | 1.04 (0.96) | 1.39 (1.12) | 1.96 (1.75) | 1.36 (1.04) | 1.83 (1.76) |
| Min. /Max. | 0.09 / 2.49 | 0.14 / 3.34 | 0.46 / 4.41 | 0.29 / 4.29 | 0.10 / 4.49 | 0.55 / 4.45 |
| | | | | | | |
| Ammonium (MARGA) mean (median) | 0.83 (0.74) | 0.74 (0.59) | 1.23 (1.10) | 1.28 (1.14) | 0.94 (0.61) | 1.38 (1.12) |
| Min. /Max. | 0.08 / 3.18 | 0.11 / 2.96 | 0.29 / 3.58 | 0.21 / 3.14 | 0.10 / 4.06 | 0.25 / 4.53 |
| | | | | | | |
| Potassium (MARGA) mean (median) | 0.14 (0.13) | 0.12 (0.11) | 0.13 (0.12) | 0.14 (0.14) | 0.12 (0.12) | 0.13 (0.12) |
| Min. /Max. | 0.1 / 0.26 | 0.10 / 0.18 | 0.10 / 0.20 | 0.11 / 0.22 | 0.10 / 0.18 | 0.10 / 0.20 |
| | | | | | | |
| Sodium (MARGA) mean (median) | 0.26 (0.17) | 0.22 (0.20) | 0.17 (0.10) | 0.10 (0.09) | 0.11 (0.10) | n.d. |
| Min. /Max. | 0.08 / 1.24 | 0.08 / 0.52 | 0.08 / 1.02 | 0.08 / 0.12 | 0.08 / 0.16 | n.d. |

260   n.d. : not detected

**Table SI-4: continued**

| cluster (hours) | C1 (307) | C2 (165) | C3 (179) | C4 (84) | C5 (88) | C6 (173) |
|---|---|---|---|---|---|---|
| region of air mass origin | West | Northwest | Southwest | South | Northeast | East |
| **Organic aerosol factors in µg m$^{-3}$** | | | | | | |
| HOA mean (median) | 0.08 (0.07) | 0.07 (0.06) | 0.11 (0.10) | 0.07 (0.05) | 0.08 (0.05) | 0.13 (0.13) |
| Min. /Max. | 0.00 / 0.44 | 0.00 / 0.26 | 0.01 / 0.39 | 0.01 /0.53 | 0.01 /0.29 | 0.01 /0.32 |
| | | | | | | |
| BBOA mean (median) | 0.37 (0.29) | 0.30 (0.25) | 0.44 (0.34) | 0.27 (0.23) | 0.26 (0.13) | 0.53 (0.53) |
| Min. /Max. | 0.00 / 2.77 | 0.03 / 1.27 | 0.06 / 1.88 | 0.09 /0.90 | 0.00 /1.43 | 0.00 /1.27 |
| | | | | | | |
| MO-OOA mean (median) | 0.65 (0.50) | 0.44 (0.27) | 1.32 (1.47) | 1.49 (1.47) | 1.15 (0.44) | 2.52 (2.47) |
| Min. /Max. | 0.00 / 3.55 | 0.02 /2.87 | 0.39 / 4.48 | 0.53 /2.53 | 0.04 /4.94 | 0.06 /5.58 |
| | | | | | | |
| LO-OOA mean (median) | 0.91 (0.74) | 0.55 (0.53) | 2.32 (1.76) | 2.54 (2.44) | 0.87 (0.85) | 1.46 (1.41) |
| Min. /Max. | 0.02 / 3.38 | 0.04 /1.94 | 0.35 / 8.15 | 0.82 /5.33 | 0.00 /3.23 | 0.26 /3.83 |
| | | | | | | |
| SV-OOA mean (median) | 0.36 (0.28) | 0.39 (0.25) | 0.87 (0.76) | 0.60 (0.53) | 0.57 (0.39) | 1.06 (0.96) |
| Min. /Max. | 0.01 / 1.37 | 0.02 / 1.94 | 0.00 / 3.12 | 0.02 /2.22 | 0.00 /4.77 | 0.02 /5.26 |
| **eBC aerosol factors in µg m$^{-3}$** | | | | | | |
| eBC-HOA mean (median) | 0.12 (0.11) | 0.13 (0.11) | 0.18 (0.15) | 0.13 (0.10) | 0.12 (0.08) | 0.22 (0.21) |
| Min. /Max. | 0.01 / 0.48 | 0.03 / 0.73 | 0.02 / 0.65 | 0.02 / 0.89) | 0.01 / 0.48 | 0.02 / 0.53 |
| | | | | | | |
| eBC-BBOA mean (median) | 0.03 (0.02) | 0.04 (0.03) | 0.05 (0.04) | 0.03 (0.02) | 0.03 (0.01) | 0.06 (0.05) |
| Min. /Max. | 0.00 /0.15 | 0.00 / 0.29 | 0.00 / 0.19 | 0.01 / 0.09 | 0.00 / 0.15 | 0.00 / 0.14 |
| | | | | | | |
| eBC-MO-OOA mean (median) | 0.19 (0.13) | 0.15 (0.10) | 0.47 (0.45) | 0.54 (0.52) | 0.38 (0.18) | 0.91 (0,89) |
| Min. /Max. | 0.00 / 0.77 | 0.00 / 1.04 | 0.00 / 1.48 | 0.18 / 0.89 | 0.01 / 1.59 | 0.11 / 1.92 |